# High and asymmetric somato-dendritic coupling of V1 layer 5 neurons independent of visual stimulation and locomotion

**Valerio Francioni[1,2], Zahid Padamsey[1], Nathalie L Rochefort[1,2]***

[1]Centre for Discovery Brain Sciences, Edinburgh Medical School, Biomedical Sciences, University of Edinburgh, Edinburgh, United Kingdom; [2]Simons Initiative for the Developing Brain, University of Edinburgh, Edinburgh, United Kingdom

**Abstract** Active dendrites impact sensory processing and behaviour. However, it remains unclear how active dendritic integration relates to somatic output in vivo. We imaged semi-simultaneously GCaMP6s signals in the soma, trunk and distal tuft dendrites of layer 5 pyramidal neurons in the awake mouse primary visual cortex. We found that apical tuft signals were dominated by widespread, highly correlated calcium transients throughout the tuft. While these signals were highly coupled to trunk and somatic transients, the frequency of calcium transients was found to decrease in a distance-dependent manner from soma to tuft. Ex vivo recordings suggest that low-frequency back-propagating action potentials underlie the distance-dependent loss of signals, while coupled somato-dendritic signals can be triggered by high-frequency somatic bursts or strong apical tuft depolarization. Visual stimulation and locomotion increased neuronal activity without affecting somato-dendritic coupling. High, asymmetric somato-dendritic coupling is therefore a widespread feature of layer 5 neurons activity in vivo.

**\*For correspondence:**
n.rochefort@ed.ac.uk

**Competing interests:** The authors declare that no competing interests exist.

## Introduction

Active dendritic conductances impact the integration of synaptic inputs and the resulting somatic output (*Grienberger et al., 2015*; *Stuart and Spruston, 2015*). Different types of active dendritic currents have been described to support local dendritic nonlinear events, primarily driven by NMDA- and voltage-gated conductances (*Schiller et al., 1997*; *Larkum et al., 1999b*). In cortical layer 5 pyramidal neurons, nonlinear dendritic spikes have been reported in the long apical dendrite and tuft, both in vitro and in vivo (*Grienberger et al., 2015*; *Stuart and Spruston, 2015*). These dendritic events are generally associated with calcium influx both locally and globally in the apical tuft (*Xu et al., 2012*; *Hill et al., 2013*; *Palmer et al., 2014*; *Cichon and Gan, 2015*; *Manita et al., 2015*; *Takahashi et al., 2016*). Additionally, the coincident occurrence of back-propagating action potentials and tuft depolarization was shown to generate widespread calcium transients in the apical tuft dendrites (*Spruston et al., 1995*; *Magee, 1997*; *Svoboda et al., 1997*; *Helmchen et al., 1999*; *Larkum et al., 1999b*; *Waters et al., 2003*; *Manita et al., 2015*).

The frequency and dynamics of local dendritic nonlinearities in vivo have been a matter of debate. Events described as local dendritically-generated spikes were reported in layer 2/3 neurons of the visual and somatosensory cortex during sensory stimulation (*Smith et al., 2013*; *Palmer et al., 2014*). In the visual cortex, it has been suggested that dendritic spikes triggered by visual input enhance somatic tuning to the preferred orientation (*Smith et al., 2013*). However, single dendrite recordings prevented conclusions about the spatial extent and the branch-specific nature of the detected events. Using calcium imaging in the mouse motor cortex, one study suggested that

individual branches of densely labelled layer 5 neurons selectively display local calcium transients correlated with specific motor outputs (*Cichon and Gan, 2015*). However, other studies reported prevalent global calcium signals throughout layer 5 apical tuft dendrites in the barrel and motor cortex (*Xu et al., 2012*; *Hill et al., 2013*). While layer 5 apical dendritic calcium transients were shown to correlate with behaviourally relevant features (*Xu et al., 2012*; *Takahashi et al., 2016*; *Peters et al., 2017*; *Ranganathan et al., 2018*; *Kerlin et al., 2019*), the coupling between apical tuft activity and somatic output remains poorly understood.

By using semi-simultaneous imaging of apical trunk and somatic GCaMP6f signals in layer 5 neurons, a recent study has shown that apical trunk dendritic calcium signals were highly correlated with somatic signals in the primary visual cortex (V1) of awake behaving mice (*Beaulieu-Laroche et al., 2019*). Layer 5 pyramidal neurons in V1 display selective responses to physical features of visual stimuli, such as the orientation and direction of movement (*Niell and Stryker, 2008*; *Kim et al., 2015*) and their activity is modulated by locomotion (*Erisken et al., 2014*; *Dadarlat and Stryker, 2017*; *Pakan et al., 2018a*). However, *Beaulieu-Laroche et al. (2019)* found that neither visual stimulation nor locomotion altered the high somato-dendritic coupling of calcium transients.

Here, we used a similar approach to image GCaMP6s calcium signals in the soma, trunk and distal tuft dendrites of layer 5 pyramidal neurons, both in darkness and during the presentation of drifting gratings, while head-fixed mice were either running or stationary. In agreement with *Beaulieu-Laroche et al. (2019)*, we found that the apical trunk dendritic calcium signals were highly correlated with somatic signals. We extended these results by showing that the whole apical tuft was highly correlated to trunk activity; the vast majority of dendritic calcium transients in the apical tuft were coincident with global events, with rare cases of branch-specific activity, limited to the smallest amplitude events. In addition, by imaging somatic and dendritic GCaMP6s signals, we found that somato-dendritic coupling was asymmetric; while almost all events observed in the tuft were also visible in the soma, around 40% of somatic events attenuated in an amplitude and distance-dependent manner from the soma to the apical tuft. Ex vivo recordings of GCaMP6s signals suggested that coupled somato-dendritic events likely reflected either strong apical tuft inputs or high frequency somatic activity, whereas attenuated events were likely caused by low frequency back-propagating action potentials. We compared GCaMP6s with GCaMP6f signals and showed that due to a reduced sensitivity, attenuation of events from soma to dendrite were underestimated with GCaMP6f (*Beaulieu-Laroche et al., 2019*). Finally, our results show that neither visual stimulation nor locomotion affected the in vivo coupling of somatic and apical tuft calcium signals.

## Results

### Highly correlated, widespread calcium signals in apical tuft dendrites of single layer 5 neurons

We imaged changes in fluorescence over time in apical dendrites of individual layer 5 pyramidal neurons sparsely labelled with either the calcium sensor GCaMP6s or GCaMP6f (*Chen et al., 2013*) in the primary visual cortex of adult mice. Using single-plane two-photon imaging (at 120 Hz), we first monitored changes of GCaMP6f signals in individual apical tuft dendrites of head-fixed, awake behaving mice that were free to run on a cylindrical treadmill (*Figure 1A*). Our results showed highly correlated calcium signals in all apical dendritic branches belonging to the same neuron and imaged in a given field of view (Pearson's correlation coefficient r = 0.92, SEM = 0.01, n = 25 fields of view recorded from 14 neurons, 6 animals) (*Figure 1B, C and D*). Given that electrical signals are known to attenuate in a distant dependent manner, we tested whether the correlation of calcium transients between apical tuft branches decreased with distance from the apical trunk (nexus). We reconstructed each individual neuron for which we imaged the apical tuft and quantified the correlation of fluorescence signals across sibling branches. We found that calcium signals in sibling branches were highly correlated (average Pearson's correlation coefficient r = 0.92) regardless of their branching order (One-way ANOVA, p=0.34) (*Figure 1E and F*). In addition to the global, widespread calcium transients, we also observed in individual spines, calcium signals that were not correlated with dendritic signals (*Figure 1—figure supplement 1*), indicating that we could resolve spine calcium signals in our experimental conditions.

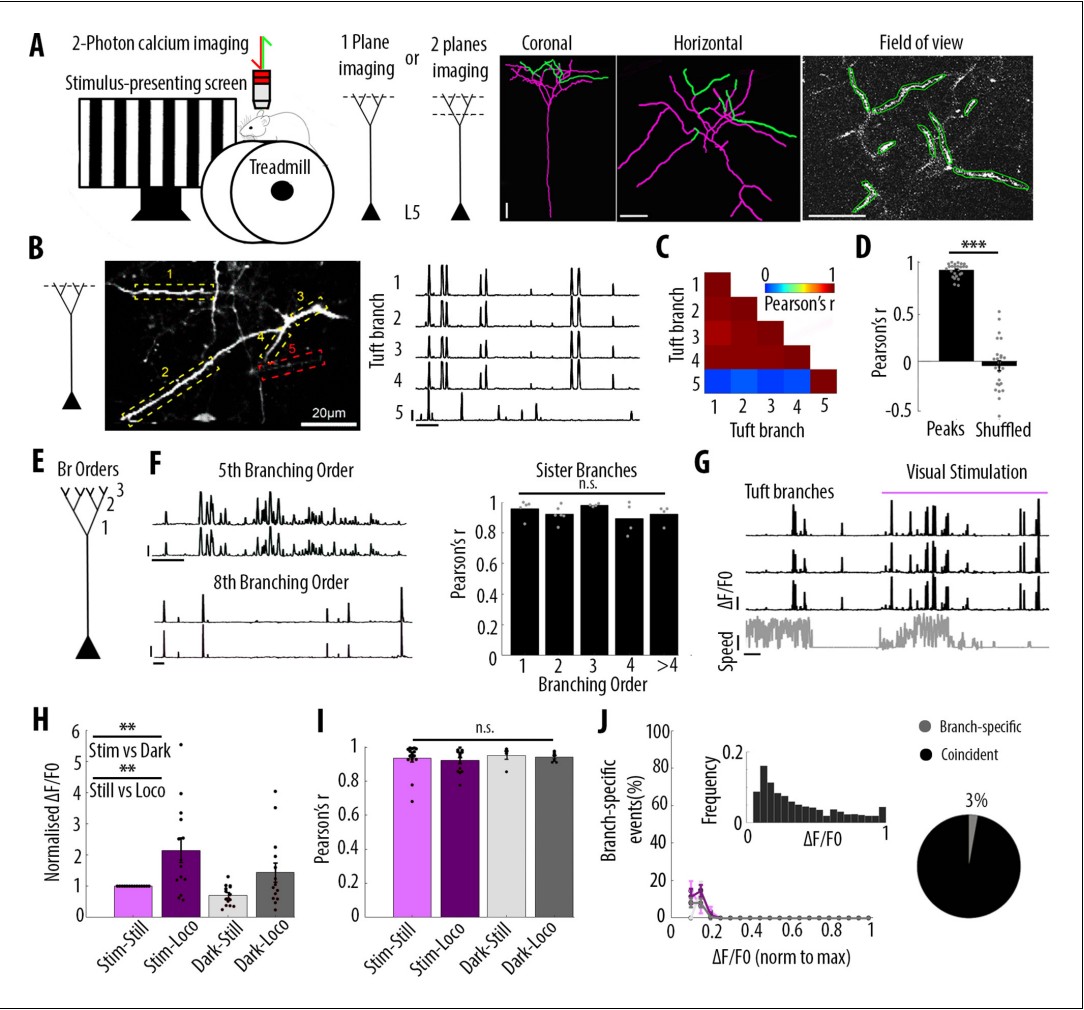

**Figure 1.** Highly correlated activity in the whole apical tuft of layer 5 pyramidal neurons. (**A**) Schematic of the methodological approach. Calcium transients were recorded either in multiple apical tuft dendrites in a single focal plane, or semi-simultaneously in different compartments of individual neurons (Soma, Trunk and Tuft). At the end of each imaging session, a z-stack was recorded for post-hoc reconstruction of each imaged neuron. (**B**) Single plane imaging of the apical tuft branches of an individual layer 5 neuron. GCaMP6f-calcium transients of the apical tuft branches belonging to one neuron are shown in traces 1 to 4 while calcium transients of a tuft branch belonging to a different neuron is shown in trace 5. Scale bars, 0.3 ΔF/F0 (normalised to max), 20 s. (**C**) Pearson's correlation matrix between the calcium transients of the branches shown in B. Branches that belong to the same neuron (branches 1 to 4, neuron 1) have a mean Pearson's correlation of 0.96, while the branch that belongs to the different neuron (branch 5, neuron 2) has a mean Pearson's correlation value of 0.13 with neuron 1's branches. (**D**) Mean Pearson's correlation value between tuft dendrites calcium transient peak amplitudes for each imaged field of view and corresponding shuffled data (Paired t-test, p=3.5e$^{-15}$; mean = 0.92 and −0.04, sem = 0.01 and 0.05 for branches and shuffled data, respectively; n = 25 fields of view; coming from 14 neurons; six animals; 70 branches). (**E**) Schematic of the definition of branching orders. (**F**) Two representative traces of sibling branches belonging to the 5$^{th}$ and 8$^{th}$ branching order and a quantification of the Pearson's correlation for sibling tuft branches of different branching orders (right panel). (For 1$^{st}$, 2$^{nd}$, 3$^{rd}$, 4$^{th}$ and more than 4$^{th}$ branching order, One-way ANOVA, p=0.34; mean = 0.96; 0.92; 0.98; 0.89; 0.92, sem = 0.02; 0.02; 0.01; 0.05; 0.03, n = 5; 6; 4; 4; 4 pairs of branches, respectively). Scale bars, 0.3 ΔF/F0 (normalised to max), 20 s. (**G**) Representative traces of 3 tuft branches belonging to the same neuron while the animal was either stationary or running (grey signal, speed) during either darkness or the presentation of drifting gratings (purple segment). Scale bars, 12 cm/s, 0.3 ΔF/F0 (normalised to max), 20 s. (**H**) Mean ΔF/F0 for apical tuft dendrites during darkness (dark) and visual stimulation (stim) while the animal was either stationary (still) or running (loco). Bar graph is normalised to visual stimulation during stationary condition. Both visual stimulation and locomotion significantly increase the mean ΔF/F0 of tuft branches without any interaction effect (Repeated Measures Two-way ANOVA on log-transformed data, p=0.005, 0.003 and 0.99 for

*Figure 1 continued*

stim, loco and interaction effects respectively, mean (normalised to condition stim/still)=1; 2.15; 0.7; 1.4, sem = 0; 0.38; 0.08 and 0.3, n = 14 neurons). (I) Pearson's correlation coefficients between tuft dendrites calcium transients of individual neurons during stim-still, stim-loco, dark-still and dark-loco conditions. (Two-way ANOVA, p=0.43, 0.62 and 0.97 for stim, loco and interaction effects; mean = 0.93; 0.92; 0.95 and 0.94, sem = 0.02; 0.02; 0.02; 0.01, n = 16; 14; 5 and 7 fields of views respectively, from 14 neurons). (J) Proportion of branch-specific events as a function of GCaMP6f calcium transients' amplitude for the four different conditions (left panel). Neither visual stimulation nor locomotion nor an interaction effect significantly increased the number of branch-specific events (Three-way ANOVA, p=0.29; 0.8 and 0.94, respectively; $p < 10^{-15}$ for event amplitude; n = 70 branches from 14 neurons; 6 animals). *Inset*, frequency distribution of the calcium transients' amplitudes detected in the apical tuft branches. *Right panel*, pie chart showing that, on average, 3% of all calcium transients were detected as branch-specific (detected in one apical tuft dendritic branch and not detected in the other imaged branches).

The online version of this article includes the following figure supplement(s) for figure 1:

**Figure supplement 1.** Single spine calcium transients in the apical tuft dendrites of layer 5 neurons.
**Figure supplement 2.** Impact of different thresholds and coincident windows on the quantification of branch-specific and compartment-specific events.

We then assessed whether these widespread, highly correlated apical tuft calcium signals were modulated by visual stimulation and locomotion. We found that both visual stimulation and locomotion increased calcium signals in the apical tuft of layer 5 neurons (mean ΔF/F0; repeated measures Two-way ANOVA, p=0.005, 0.003 and 0.99 for visual stimulation, locomotion and interaction between both conditions, respectively; n = 14 neurons) (*Figure 1G and H*). However, the correlation of calcium signals between all imaged dendritic branches remained high (r ≥ 0.92) during periods of darkness and visual stimulation as well as during stationary and locomotion periods, without a significant difference across conditions (Two-way ANOVA, p=0.43, 0.62 and 0.97 for visual stimulation, locomotion and interaction between both conditions, respectively) (*Figure 1I*).

In line with these findings, we found that branch-specific activity in the apical tuft was rare. We quantified the amount of branch-specific activity in all pairs of branches belonging to the same neuron as the proportion of calcium transients present in one branch and absent in the other. Among the 70 imaged branches across 14 neurons, branch-specific calcium transients represented less than 3% of the total number of transients. Among all imaged calcium transients, these local signals were dominated by calcium transients of the smallest amplitudes (*Figure 1J*). In addition, neither visual stimulation nor locomotion significantly affected this small proportion of branch-specific calcium signals (Three-way ANOVA, $p < 10^{-15}$ for event amplitude and p=0.29; 0.8 and 0.94, for visual stimulation, locomotion and interaction between both conditions, respectively; no other interaction effect was statistically significant) (*Figure 1J*). We tested the robustness of our quantification by using different thresholds for the detection of individual calcium transients (*Figure 1—figure supplement 2A and B*). We found that lowering the threshold by 30% in one of the two pair branches reduced the proportion of branch-specific events from 3.8% to 0.8%, suggesting that most of these already rare events were near detection threshold (*Figure 1—figure supplement 2B*). We also confirmed that our results were not affected by the time window that we used to detect coincident events in pairs of dendritic branches (*Figure 1—figure supplement 2D and E*).

Altogether, our results show that apical dendritic calcium signals in V1 layer 5 neurons are almost exclusively dominated by highly correlated calcium transients in the whole apical tuft regardless of the dendritic branching order. Neither visual stimulation nor locomotion modulated this high correlation between apical tuft dendritic calcium signals.

## Calcium signals are highly correlated throughout all compartments of individual layer 5 neurons

We then assessed how global, widespread apical tuft calcium signals were coupled across neuronal compartments. For this, we semi-simultaneously imaged GCaMP6s calcium signals at two focal planes, separated by 170 μm in depth, in four different pairs of neuronal compartments from the soma to the apical tuft: Soma-Trunk, proximal Trunk-distal Trunk, distal Trunk-apical Tuft, proximal Tuft-distal Tuft (*Figure 2A and B*). We used GCaMP6s which has higher signal amplitude and signal-to-noise ratio than GCaMP6f, in order to maximise our ability to detect somatic signals (*Chen et al.,*

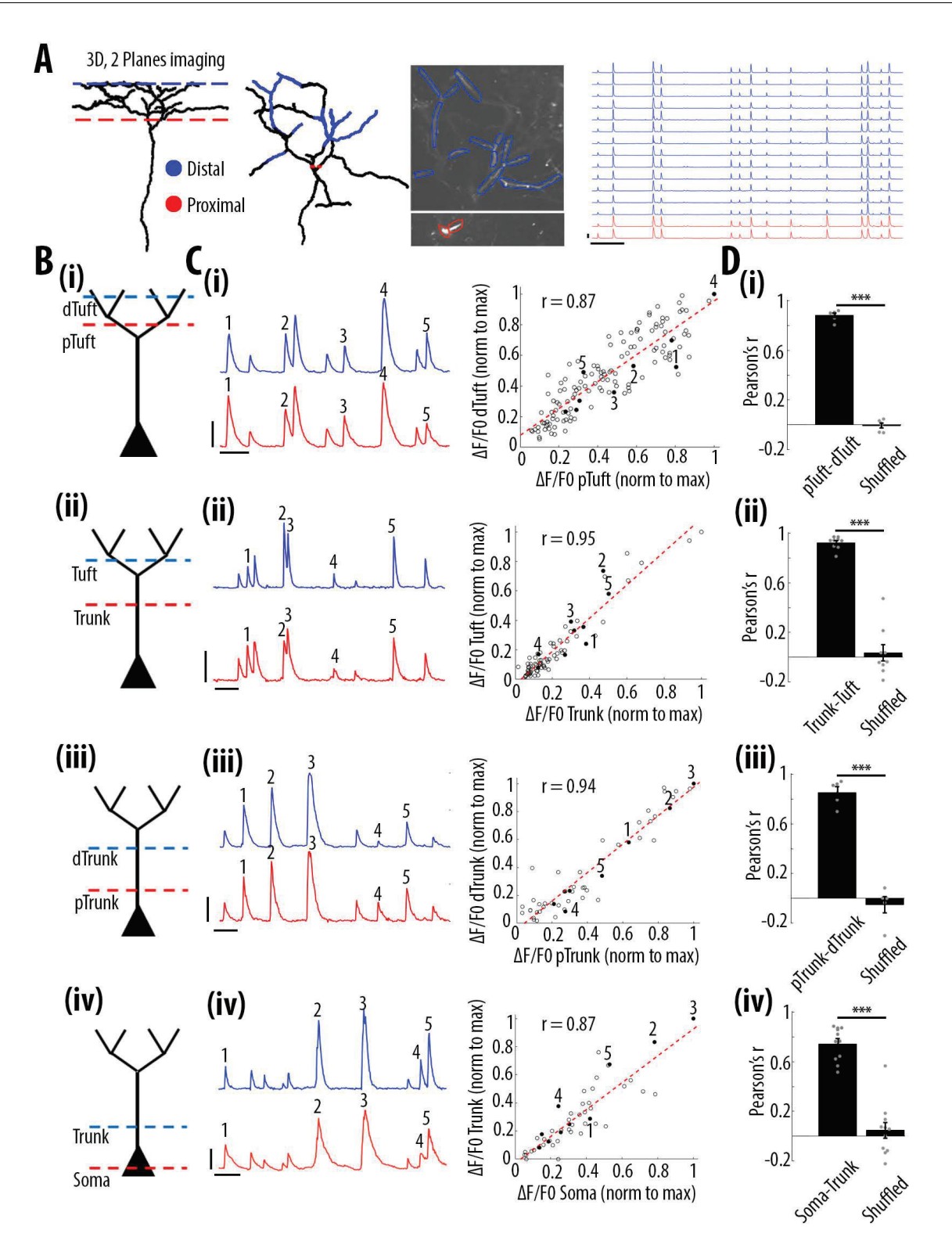

**Figure 2.** High correlation of calcium transients between neuronal compartments (soma, trunk, apical tuft) of individual layer 5 neurons. (**A**) Anatomical reconstruction of an individual GCaMP6s-labeled layer 5 pyramidal neuron imaged at two focal planes semi-simultaneously (red dotted lines, proximal; blue, distal). *Left panel*, coronal and horizontal views of the same imaged neuron. *Right panel*, two-photon image of dendritic branches highlighted in red and blue in the horizontal view of the anatomical reconstruction. Example ΔF/F0 traces of highly correlated calcium transients from dendritic

*Figure 2 continued on next page*

*Figure 2 continued*

branches indicated in red and blue. (**B**) Schemata of the compartments imaged semi-simultaneously, (**i**) proximal tuft-distal tuft, (**ii**) trunk-tuft, (**iii**), trunk-trunk, (**iv**) soma-trunk. pTrunk, dTrunk and pTuft, dTuft indicate proximal and distal portions of the trunk and the apical tuft, respectively. (**C**) *Left panel*, representative GCaMP6s transients imaged in two neuronal compartments semi-simultaneously as shown in B. *Right panel*, scatter plot of peak amplitudes of individual calcium transients in proximal and distal compartments imaged semi-simultaneously, in one example individual neuron. Each dot represents a calcium transient. Peak amplitudes were normalized to the maximum amplitude in each compartment. Filled dots correspond to the transients indicated by numbers in the left panel. Red dotted line indicates the best fit (least square). Pearson's correlation values (**r**) are indicated for each example pair of neuronal compartments. Scale bars 0.3 ΔF/F0 (normalised to max), 10 s. (**D**) Pearson's correlation values for each pair of compartments imaged semi-simultaneously and corresponding shuffled data (Paired t-test, (i)pTuft-dTuft, $p=1.6e^{-6}$, mean = 0.88; −0.01, sem = 0.02; 0.02, n = 6 pairs; (ii)Trunk-Tuft, $p=8e^{-7}$, mean = 0.92; 0.04, sem = 0.02; 0.07, n = 9; (iii) pTrunk-dTrunk, $p=4.4e^{-4}$, mean = 0.85; −0.05, sem = 0.05; 0.07, n = 5; (iv) Soma-Trunk, $p=4.4e^{-6}$, mean = 0.74; 0.05, sem = 0.04; 0.06, n = 11).

The online version of this article includes the following figure supplement(s) for figure 2:

**Figure supplement 1.** Scatter plots of peak amplitudes of individual calcium transients in all proximal and distal compartments imaged semi-simultaneously.

**Figure supplement 2.** Most dendritic events detected as branch-specific were also detected in the trunk.

*2013*). We found that calcium transients simultaneously imaged in each pair of proximal and distal compartments were highly correlated (*Figure 2A-D*). Since the rise and decay kinetics of calcium events were slower in the soma than in the dendrites, we performed our quantification on individual peaks of calcium transients (see Materials and methods). For each pair of compartments, we found that the average Pearson's correlation value between the peak amplitude of individual transients was 0.88 between the proximal and distal parts of the apical tuft, 0.92 between the trunk and the apical tuft and 0.85 between the proximal and distal trunk (*Figure 2D* and *Figure 2—figure supplement 1*). The lowest correlation value was found between the soma and the proximal trunk (0.74, *Figure 2D*). These results also confirmed the high correlation of calcium transients throughout the apical tuft (*Figure 2A*, 2D(i)), that we found in our single-plane imaging experiments (*Figure 1D and F*). We checked whether the small proportion of events (3%) that were detected as branch-specific in apical tuft dendrites were also found in the apical trunk. On average, we found that 60% of the events detected as branch-specific were also found in the trunk (*Figure 2—figure supplement 2*).

Altogether, these results show that calcium transient amplitudes were highly correlated from the soma to the distal apical tuft of V1 layer 5 neurons.

## Frequency of calcium transients decreases in a distance- and amplitude-dependent manner from soma to apical tuft

We then tested whether the high correlation of calcium transient amplitudes from the soma to the apical tuft was associated with distance-dependent changes in the frequency of these events. We quantified the frequency of GCaMP6s transients in each pair of proximal and distal compartments imaged semi-simultaneously (*Figure 3A*). We found that the frequency of calcium transients decreased from proximal to distal compartments (Paired t-test, $p=1.1e^{-6}$, n = 31 pairs of compartments from 19 neurons) by an average of 14% from soma to proximal trunk, 8% from proximal to distal trunk, 24% from distal trunk to apical tuft and 22% from proximal tuft to distal tuft (*Figure 3B*). From these proportions, we estimated a decrease of about 40% of calcium transients from soma to the distal part of the apical tuft (*Figure 3—figure supplement 1A and B*). This result was confirmed by a second data set, in which somatic and apical tuft calcium transients were imaged independently (at 120 Hz) in individual layer 5 neurons. We found that the mean frequency of calcium transients in the apical tuft corresponded to 62% of the frequency of events in the corresponding soma (n = 13 neurons *Figure 3—figure supplement 1C*).

We then checked whether this decrease in frequency depended on the amplitude of the calcium transients. We calculated the percentage of compartment-specific calcium transients as a function of their amplitude (*Figure 3C*). Our results show that smaller amplitude calcium transients in the proximal compartment were more likely to attenuate below detection level in the distal compartment compared to larger amplitudes events (red trace in *Figure 3C*). The vast majority of compartment-specific events were dominated by calcium transients in proximal compartments that were not detected in distal ones (red trace in *Figure 3C*), while only few calcium transients were found in distal compartments and not in the corresponding proximal compartment (blue trace in *Figure 3C*)

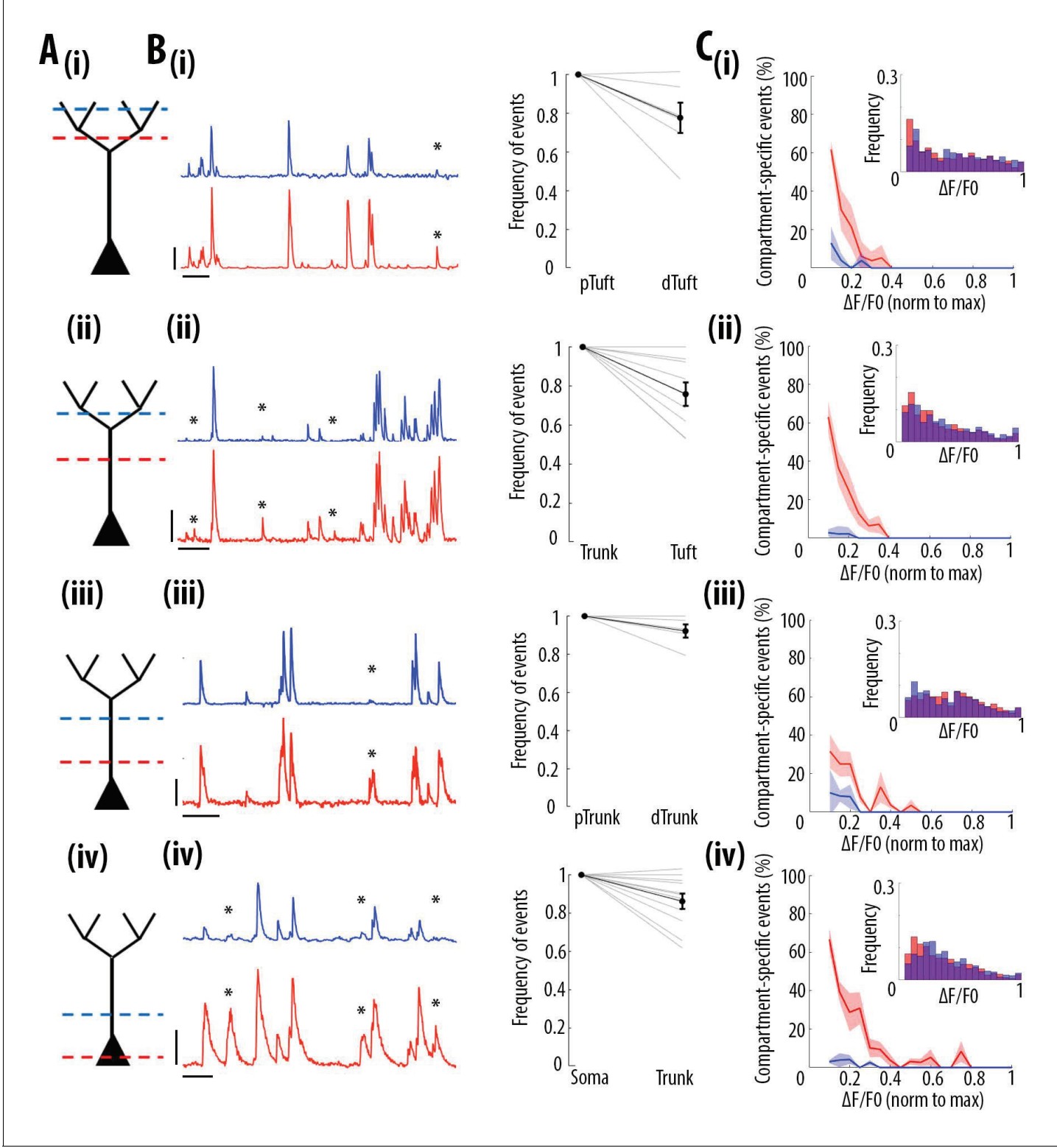

**Figure 3.** Frequency of calcium transients decreases in a distance and amplitude-dependent dependent manner from the soma to the apical tuft. (**A**) Schemata of the neuronal compartments imaged simultaneously in individual layer 5 neurons. (**B**) *Left panel*, representative ΔF/F0 traces of GCaMP6s calcium transients imaged semi-simultaneously in two different compartments as indicated in panel A. Asterisks indicate calcium transients detected in the proximal compartment (red trace) and not detected in the distal one (blue trace). Scale bars 0.25 ΔF/F0 (normalised to max), 20 s. *Right panel*, frequency of detected calcium transients, normalised to the proximal compartment. Individual lines represent individual neurons. Error bar: SEM. (**C**)

*Figure 3 continued on next page*

*Figure 3 continued*

Proportion of compartment-specific events as a function of calcium transients' amplitude. In red, proportion of events only detected in the proximal compartment. In blue, proportion of events only detected in the distal compartment. Thick line represents the weighted mean proportion. Shaded area represents the weighted sem for each bin (0.05). *Upper right panel*, frequency histogram of calcium transient peak amplitudes detected in the proximal (red) and distal (blue) compartments. Peak amplitudes were normalized to the maximum amplitude in each compartment. For all compartments, event amplitude, compartment (proximal vs distal) and an interaction between these two factors significantly affected the proportion of compartment-specific events (Two-way ANOVA, (i)pTuft-dTuft, $p<10^{-15}$, $p=7.8\times10^{-5}$, $p=6.2\times10^{-8}$, for event amplitude, proximal versus distal compartment and interaction between amplitude and compartment, respectively, n = 6 pairs; (ii)Trunk-Tuft, $p<10^{-15}$, $p=3.4\times10^{-9}$, $p<10^{-15}$, n = 9; (iii) pTrunk-dTrunk, $p=1.3\times10^{-13}$, $p=2.4\times10^{-4}$, $p=7\times10^{-3}$, n = 5; (iv) Soma-Trunk, $p<10^{-15}$, $p<10^{-15}$ and $p<10^{-15}$, n = 11).

The online version of this article includes the following figure supplement(s) for figure 3:

**Figure supplement 1.** Frequency decrease of calcium transients from soma to apical tuft.

(see also *Video 1* and *Video 2*). As a result, compartment-specific events were dominated by small amplitude calcium transients that attenuated from proximal to distal compartments (Two-way ANOVA, $p<10^{-15}$, $p<10^{-15}$, $p<10^{-15}$, for compartment (proximal vs distal), amplitude, and interaction effect, respectively; Figure 5A). Our results were robust to the threshold (+/- 30%) that was used for detecting calcium transients (*Figure 1—figure supplement 2A and C*). Our results were also robust to the time window that was used to detect simultaneous events in different compartments (*Figure 1—figure supplement 2D and F*). Notably, we also observed relatively large calcium transients in soma and proximal trunk that were not detected in the distal compartment (see examples in *Figure 3B (iii) and (iv)* and percentage of compartment-specific event in *Figure 3C (iii) and (iv)* for ΔF/F0 >0.3), suggesting the presence of active mechanisms inhibiting calcium transients along the apical trunk.

These results indicate that almost all distal events were found in proximal compartments, while at least 40% of somatic transients attenuate in a distance-dependent manner along the apical trunk and distal dendrites. Thus, we found that somato-dendritic coupling in layer 5 neurons was strong, but asymmetric.

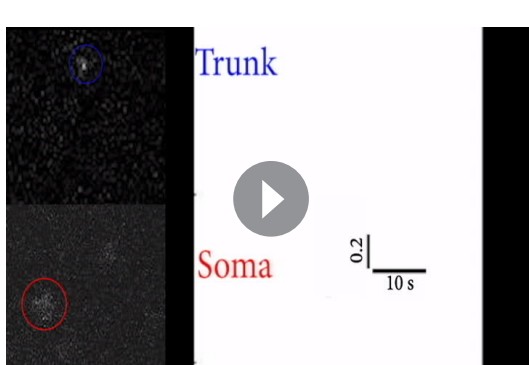

**Video 1.** Two-photon imaging movie showing calcium signals detected in the soma and not detected in the corresponding trunk of an individual layer 5 pyramidal neuron in V1. *Left panel,* two-photon calcium imaging (raw data) for a pair of neuronal compartments (Soma-Trunk) imaged semi-simultaneously. Both the soma (*lower quadrant,* circled in red) and the corresponding trunk (*upper quadrant,* circled in blue) of an individual layer 5 pyramidal neuron are shown. *Right panel,* GCaMP6s signals of the two compartments (red for Soma, blue for Trunk) shown in the video on the left panel. Data were acquired at 4.8 Hz per plane. The two fields of view were 170 μm apart in the coronal plane. Scale bars: 10 s and 0.2 ΔF/F0 (normalised to max).
https://elifesciences.org/articles/49145#video1

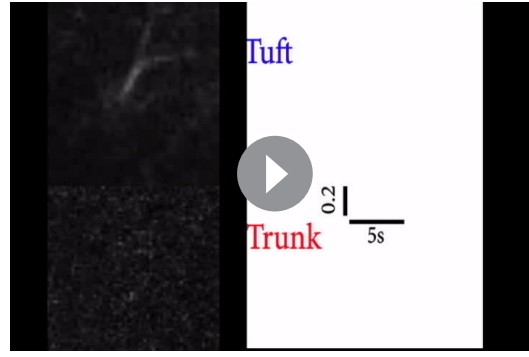

**Video 2.** Two-photon imaging movie showing calcium signals detected in the apical trunk and not detected in the corresponding tuft of an individual layer 5 pyramidal neuron in V1. *Left panel,* two-photon calcium imaging (raw data) for a pair of neuronal compartments (Trunk-Tuft) imaged semi-simultaneously. Both the apical trunk (*lower quadrant,* circled in red) and the corresponding tuft (*upper quadrant,* circled in blue, three branches) of an individual layer 5 pyramidal neuron are shown. *Right panel,* GCaMP6s signals of the two compartments (red for Trunk, blue for Tuft (upper right branch)) shown in the video on the left panel. Data were acquired at 4.8 Hz per plane. The two fields of view were 170 μm apart in the coronal plane. Scale bars: 5 s and 0.2 ΔF/F0 (normalised to max).
https://elifesciences.org/articles/49145#video2

## Ex vivo calibration of GCaMP6s and GCaMP6f signals in layer 5 soma and apical tuft dendrites

We next sought to characterize the electrophysiological activity underlying our observations of strong, but asymmetric, somato-dendritic coupling in vivo. We reasoned that neuronal spiking driven by stimulation of either the soma or the apical tuft would differentially impact calcium event amplitudes in proximal and distal neuronal compartments. We examined this ex vivo by combining calcium imaging and whole-cell recordings in acute cortical slices from mouse V1. We imaged the soma and distal apical dendrite (nexus) of GCaMP6s- and GCaMP6f- expressing layer 5 neurons whilst driving neuronal spiking either by somatic current injections (somatic stimulation) or L1 electrical stimulation (dendritic stimulation) (*Figure 4A*). Somatic and dendritic stimulation consisted of a train of 10 suprathreshold stimuli delivered at 5–200 Hz; each stimulus reliably triggered single action potentials (*Figure 4B*). Action potentials evoked by dendritic stimulation were accompanied by an enhanced afterdepolarization in the soma, characteristic of distal dendritic electrogenesis (*Larkum et al., 1999a*; *Shai et al., 2015*).

For somatic stimulation, we found that we could robustly detect somatic calcium signals with firing frequencies as low as 5 Hz with GCaMP6s (*Figure 4E, G*), and as low as 25 Hz with GCaMP6f (*Figure 4F, H*); we could not reliably detect calcium events associated with single action potentials with either indicator ($\Delta$F/F0 peak for GCaMP6s: mean = 0.02, sem = 0.01, t-test vs null, p=0.07, n = 10 neurons; $\Delta$F/F0 peak for GCaMP6f: mean = $-0.02$, sem = 0.03, t-test vs null, p=0.39, n = 9 neurons). Somatic signal amplitude increased monotonically with stimulation frequency for both indicators, with a plateau at 100–200 Hz. By contrast, in the distal dendrites, reliable signals for both indicators were only detected when somatically-driven spiking exceeded 50 Hz (*Figure 4E–H*), likely reflecting the critical frequency (50–100 Hz) of back-propagating action potentials required to trigger distal dendritic calcium spikes (*Larkum et al., 1999a*; *Shai et al., 2015*). For dendritic stimulation, calcium signals were reliably detected in both the soma and distal dendrites. With GCaMP6s, this was true for all frequencies tested (*Figure 4C, E, G*). With GCaMP6f, somatic signals could not be observed at lower stimulation frequencies (5 Hz) (*Figure 4D, F, H*).

Our findings suggest that: 1) somatically-triggered back-propagating action potentials below the critical frequency (<50–100 Hz) may underlie the distant-dependent loss of dendritic calcium signals we observed in vivo; 2) spiking driven by strong apical tuft input or high-frequency spiking driven by somatic depolarization may underlie the strong somato-dendritic coupling detected in vivo.

Notably, compared to GCaMP6s, GCaMP6f had a reduced sensitivity for detecting calcium signals, specifically in the soma during low frequency spiking driven either by somatic or dendritic stimulation (*Figure 4E–H*). Consequently, as compared to GCaMP6s, attenuation of events from soma to dendrite was underestimated whereas attenuation of events from dendrite to soma was overestimated with GCaMP6f (*Figure 4G vs 4H*). With GCaMP6s, by contrast, attenuation of signals was asymmetric; we only found attenuation from soma to distal dendrites, as observed in vivo (*Figure 4G*).

We further examined GCaMP6s signals across a number of dendritic stimulation intensities (25–100% of somatic spike threshold) and frequencies (5–200 Hz) (*Figure 4—figure supplement 1C*) allowing for analysis of ex vivo somato-dendritic coupling across a range of stimulus-evoked depolarizations. We could not detect any global calcium signals in the apical tuft when L1 stimulation failed to evoke a somatic action potential, indicating a strong coupling between global calcium events and somatic spiking in our experimental conditions (*Figure 4—figure supplement 1D*). Thus, no calcium signal was detected in either compartment (soma and apical tuft) in trials with no action potential. When L1 stimulation only evoked a single action potential, apical tuft calcium signals were observed in 60% of the trials; however, these events were below the detection threshold of GCaMP6s in the soma (*Figure 4—figure supplement 1F*). Consequently, under these conditions, somato-dendritic coupling was low. Provided L1 stimulation evoked at least two action potentials, calcium signals in both apical tuft and soma were robustly detected (*Figure 4—figure supplement 1H*). Across stimulation parameters, both somatic and dendritic calcium transients increased with the number of action potentials evoked by L1 stimulation (*Figure 4—figure supplement 1D*). Collectively, our findings reveal that evoked apical tuft calcium events are strongly coupled to somatic spiking, and that this coupling can be detected by GCaMP6s imaging provided that somatic output is above the limits of detection ($\geq$2 action potentials ex vivo).

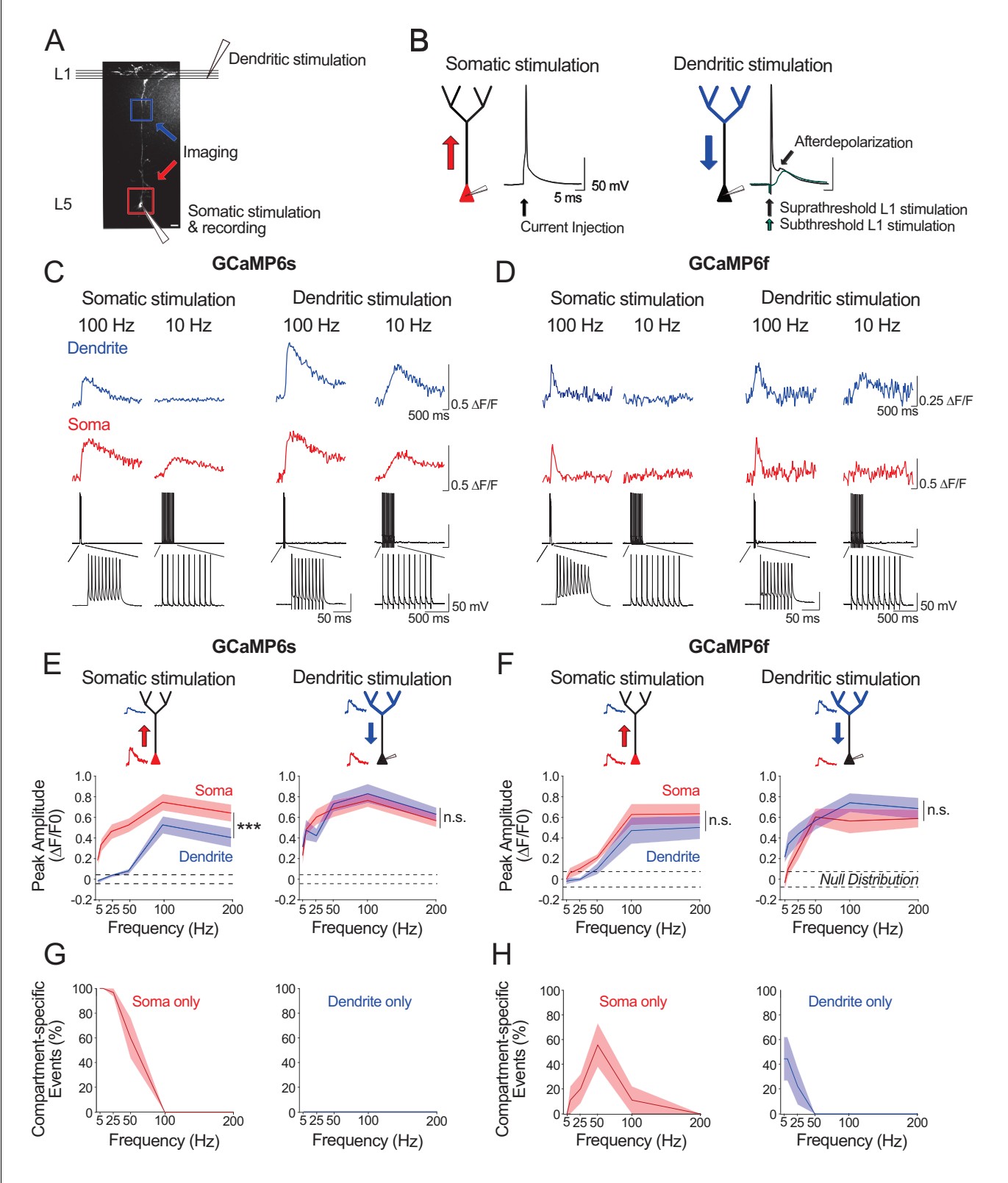

**Figure 4.** Ex vivo calibration of GCaMP6s and GCaMP6f signals in soma and apical tuft dendrites. (**A**) Experimental schemata. Layer 5 pyramidal neurons expressing GCaMP6s or GCaMP6f were recorded using whole-cell patch clamp in acute cortical slices of the primary visual cortex. An example neuron is shown (scale bar = 20 μm). The soma and distal apical dendrite (nexus) were imaged during neuronal spiking, driven either by suprathreshold somatic current injection (somatic stimulation) or layer 1 (**L1**) stimulation (dendritic stimulation), respectively. Somatic and dendritic stimulation consisted

*Figure 4 continued on next page*

*Figure 4 continued*

of a train of 10 pulses delivered at 5–200 Hz. (B) Examples of action potentials driven by somatic and dendritic stimulation. Action potentials triggered by dendritic stimulation were additionally accompanied by an enhanced afterdepolarization reminiscent of a dendritic spike, which had a faster rise time than subthreshold EPSPs and were absent from somatically-evoked action potentials. (C, D) Example calcium transients imaged in the soma and apical dendrite during somatic and dendritic stimulation in a L5 neuron expressing GCaMP6s (C) and GCaMP6f (D). Concurrent somatic electrophysiological recordings are shown below each transient. Negative deflections in electrophysiological traces are stimulation artefacts that have been cut for clarity. (E, F) Average peak amplitude of calcium transients recorded in the soma (red) and dendrites (blue) during somatic and dendritic stimulation for GCaMP6s (E) and GCaMP6f (F). Null distribution of peak amplitudes from sham stimulation trials are shown (dashed horizontal lines). (G, H) Proportion of compartment-specific events detected either in the soma or dendrite during somatic or dendritic stimulation for GCaMP6s (G) and GCaMP6f (H). Shaded areas represent S.E.M. Significance was assessed with two way (E, F) and one way (G, H) repeated measures ANOVA with post-hoc Sidak tests. Asterisks (***) in (E,F) denote factor (soma vs dendrite) significance of $p<0.0001$; n.s. denotes non-significance; significance is not shown for other comparisons but is listed below. For somatic stimulation in (E): soma vs. dendrite: $p<0.0001$ for all frequencies; soma vs. null: $p<0.0001$ for all frequencies; dendrite vs null; $p<0.0001$ for frequencies $> 50$ Hz, $p>0.44$ for frequencies $\leq 50$ Hz. For dendritic stimulation in (E): soma vs. dendrite: $p>0.11$ for all frequencies; soma vs null: $p<0.0001$ for all frequencies; dendrite vs null: $p<0.0001$ for all frequencies. For somatic stimulation in (F): soma vs. dendrite: $p>0.49$ for$>25$ Hz, $p=0.074$ for 10 Hz, $p=0.054$ for 5 Hz; soma vs. null: $p<0.0001$ at$>100$ Hz, $p=0.09$ at 50 Hz, $p>0.7$ at$<50$ Hz; dendrite vs. null: $p<0.0001$ at$>100$ Hz; $p>0.52$ at$<50$ Hz. For dendritic stimulation in (F): soma vs. dendrite: $p>0.32$ for all frequencies; soma vs. null: $p<0.0001$ at$>50$ Hz, $p<0.01$ at 25 Hz, $p>0.47$ at$<10$ Hz; dendrite vs. null: $p<0.0001$ at$>25$ Hz, $p<0.002$ at 10 Hz, $p=0.11$ at 5 Hz. For somatic stimulation in (G): soma vs null: $p<0.0001$ for$<50$ Hz, $p>0.44$ for$>100$ Hz. For dendritic stimulation in (G): dendrite vs null: $p>0.11$ for all frequencies. For somatic stimulation in (H): soma vs null: $p<0.0001$ for 50 Hz; $p>0.12$ for all other frequencies. For dendritic stimulation in (H): dendrite vs null: $p<0.02$ for 5 Hz, $p=0.10$ for 10 Hz, $p>0.48$ for$>25$ Hz. GCaMP6s: n = 10 cells from 4 animals except for 5 Hz stimulation, where n = 5 cells from 3 animals; GCaMP6f: n = 9 cells from 3 animals.

The online version of this article includes the following figure supplement(s) for figure 4:

**Figure supplement 1.** Ex vivo calibration of GCaMP6s in soma and apical tuft dendrites across subthreshold and suprathreshold dendritic stimulation parameters.

## Visual stimulation and locomotion do not alter the coupling of calcium signals between neuronal compartments from soma to apical tuft

We then tested whether visual stimulation and locomotion altered the coupling between the different compartments of individual layer 5 neurons. For each condition, we plotted the percentage of compartment-specific calcium transients as a function of their amplitude. We found that in all four conditions compartment-specific events were dominated by small amplitude calcium transients in proximal compartments: there was no significant difference between darkness and visual stimulation periods, both during stationary and locomotion periods (*Figure 5B*; Three-way ANOVA, $p=0.69$, $p=0.21$ and $p=0.64$ for visual stimulation, locomotion, and interaction respectively; $p<10^{-15}$ for event amplitude; no other interaction was found to be statistically significant). Similarly, we found no significant difference between behavioural states defined by stationary, locomotion and transition from still to running periods (see Materials and methods), either during visual stimulation or in darkness (*Figure 5—figure supplement 1A*). These results indicate that the relationship between calcium transient amplitude and percentage of attenuated events from soma to apical tuft remained unchanged by visual stimulation and behavioural-state changes. In addition, we found that, on average, the percentage of coincident calcium signals across compartments of individual neurons was not significantly modified either by visual stimulation or behavioural state changes (n = 19 neurons) (*Figure 5—figure supplement 1B* and *Figure 5—figure supplement 2*).

Even though smaller events were attenuated with similar probabilities across the four conditions, calcium transients that do propagate through different neuronal compartments, may be amplified or attenuated in a condition-dependent manner (*Figure 5C*). To capture these nonlinearities, we plotted the amplitude of each detected transient in each compartment and calculated a residual value as the distance from the linear robust regression fit (*Figure 5C*). We then plotted the cumulative distribution of the residual value for every calcium transient in all 4 pairs of imaged compartments from soma to apical tuft (*Figure 5D*). We found that the distribution of the residual values across neuronal compartments were neither significantly affected by locomotion nor by visual stimulation (Three-way ANOVA, $p=0.96$, $p=0.23$ and $p=0.91$ for visual stimulation, locomotion and interaction effect, respectively; $p=0.86$ for different compartments; no other interaction effect was found to be significant; n = 11 Soma–Trunk pairs; n = 5 pTrunk–dTrunk; n = 9 Trunk–Tuft; n = 6 pTuft-dTuft from 19 neurons). These results indicate that, while locomotion and visual stimulation increase the activity of layer 5 pyramidal neurons, the relationship between the amplitude of calcium transients and the

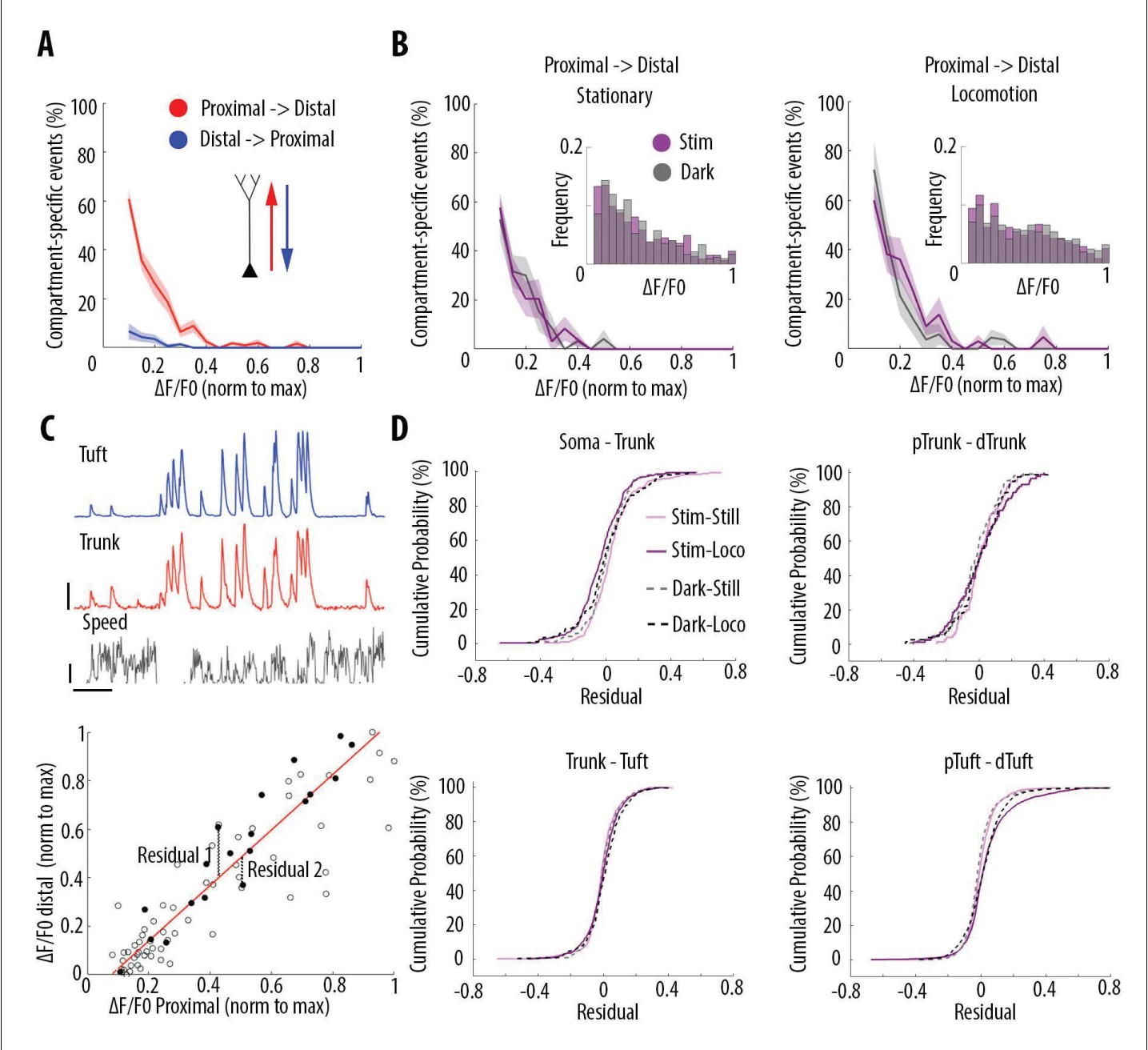

**Figure 5.** Locomotion and visual stimulation do not alter the relationship between somatic and dendritic calcium transients in layer 5 pyramidal neurons. (**A**) Mean proportion of compartment-specific calcium transients as a function of calcium transient amplitude (average of data points from *Figure 3C* (i–iv)). In red, proportion of events only detected in the proximal compartment. In blue, proportion of events only detected in the distal compartment. Thick line and shaded area: weighted mean and sem for each bin (0.05), respectively. Event amplitude, compartment (proximal vs distal) and an interaction between these two factors significantly affected the proportion of compartment-specific events (Two-way ANOVA, $p<10^{-15}$, $p<10^{-15}$, $p<10^{-15}$, n = 31 pairs of compartments from 19 neurons). (**B**) Proportion of compartment-specific events detected in the proximal compartment and not in the distal one, as a function of calcium transient amplitude, during periods of darkness (grey) and visual stimulation with drifting gratings (purple), while the animals were either stationary (left panel) or running (right panel). *Upper right panel*, frequency histogram of calcium transient peak amplitudes detected in proximal compartments during darkness (grey) and visual stimulation (purple). Peak amplitudes were normalized to the maximum amplitude in each compartment. Neither visual stimulation nor locomotion altered the function relating calcium transients amplitude with the proportion of compartment-specific events (Three-way ANOVA, p=0.69, p=0.21 and p=0.64 for visual stimulation, locomotion, and interaction respectively; $p<10^{-15}$ for event amplitude. No other interaction was found to be statistically significant). (**C**) *Upper panel*, example traces of a proximal (trunk, red) and distal (tuft, blue) compartment imaged semi-simultaneously, during stationary and locomotion periods (black trace, speed). Scale bars, 0.25 ΔF/F0 (normalised to max), 12 cm/s, 20 s. *Lower panel*, example scatter plot of calcium transients' peak amplitudes imaged in a pair of neuronal

*Figure 5 continued on next page*

*Figure 5 continued*

compartments (trunk-tuft). Each dot represents an individual calcium transient. Filled dots correspond to the transients shown in the upper panel. The red line represents the robust linear regression fit. For each transient, a residual from the robust linear regression was calculated. (D) Cumulative distributions of the residuals calculated for each pair of compartments and for each condition: visual stimulation (pink, stim), darkness (grey, dark), stationary (still) and locomotion (loco) periods. No significant difference was found between any condition: Three-way ANOVA, p=0.96, p=0.23 and p=0.91 for visual stimulation, locomotion and interaction effect, respectively; p=0.86 for different neuronal compartments. No other interaction effect was found to be significant. n = 31 compartments (n = 11 Soma-Trunk; n = 5 pTrunk–dTrunk; n = 9 Trunk–Tuft; n = 6 pTuft-dTuft from 19 neurons). The online version of this article includes the following figure supplement(s) for figure 5:

**Figure supplement 1.** Behavioural-state transitions between stationary and locomotion do not alter the relationship between somatic and dendritic calcium transients.

**Figure supplement 2.** Proportion of coincident calcium transients in proximal and distal compartments imaged semi-simultaneously during the different behavioural conditions: visual stimulation (stim), darkness (dark), stationary (still) and locomotion (loco) periods.

probability of coupled events across compartments remains unchanged. Independent of condition, single calcium transients of a given amplitude have the same probability of being compartment-specific.

Since layer 5 neurons in V1 respond selectively to drifting grating orientations and that dendritic spikes triggered by visual input were suggested to enhance somatic tuning to the preferred orientation (*Smith et al., 2013*), we quantified the impact of grating orientation on the correlation of calcium transients throughout neuronal compartments. In the apical tuft, we found that the selectivity of the responses to the drifting gratings orientation, quantified by an orientation selectivity index (OSI), did not affect the high correlation of calcium transients between tuft dendrites (*Figure 6A–D*). We then compared orientation selective responses during stationary and locomotion periods both in apical tufts and corresponding somata (n = 15 neurons) (*Figure 6E and F*). We found that locomotion similarly increased calcium transient amplitudes both in apical tuft and soma, without a significant difference between compartments (*Figure 6G*, Paired t-test, p=0.38), both during the presentation of preferred and non-preferred orientations (*Figure 6H*, Repeated measures Two-way ANOVA on log-transformed data, p=0.07, 0.71 and 0.28 for stimulus-type, compartment and interaction effects respectively). As a consequence, the OSI of somatic and apical tuft responses remained unchanged during both stationary and locomotion periods (*Figure 6I*, Repeated measures Two-way ANOVA, p=0.49, p=0.42 and p=0.45, for the effects of locomotion, neuronal compartment and interaction, respectively). In line with these results, we found that the preferred orientation was similar throughout tuft dendritic branches and neuronal compartments of individual neurons, both during stationary and locomotion periods (*Figure 6J and K*). Finally, the Pearson's correlation between apical tuft branches calcium signals and across neuronal compartments remained high and unchanged regardless of the gratings orientation (*Figure 6L and M*).

Altogether, these results indicate that calcium signals in individual layer 5 pyramidal neurons are highly correlated throughout apical tuft branches and neuronal compartments, and that this high somato-dendritic coupling remains unchanged by visual stimulation and locomotion.

## Discussion

Our results show that GCaMP6f and GCaMP6s calcium transients are highly correlated in apical tuft dendrites of individual layer 5 neurons in the primary visual cortex of awake behaving mice; branch-specific calcium transients were rare and limited to small amplitude transients. GCaMP6s calcium transients were also found to be highly correlated throughout all compartments of individual neurons (soma, trunk, tuft). However, their frequency was found to decrease in a distance-dependent manner from soma to apical tuft. Whereas almost all transients observed in the tuft were present in proximal compartments, the frequency of calcium transients in apical tuft was about 60% of their frequency in the corresponding soma. Ex vivo experiments suggest that attenuated dendritic signals were likely associated with low frequency back-propagating action potentials, whereas highly correlated somato-dendritic signals were likely associated with high frequency somatic activity, or strong apical tuft inputs. Neither visual stimulation nor locomotion altered either the coupling between neuronal compartments from soma to apical tuft, nor the proportion of tuft branch-specific events.

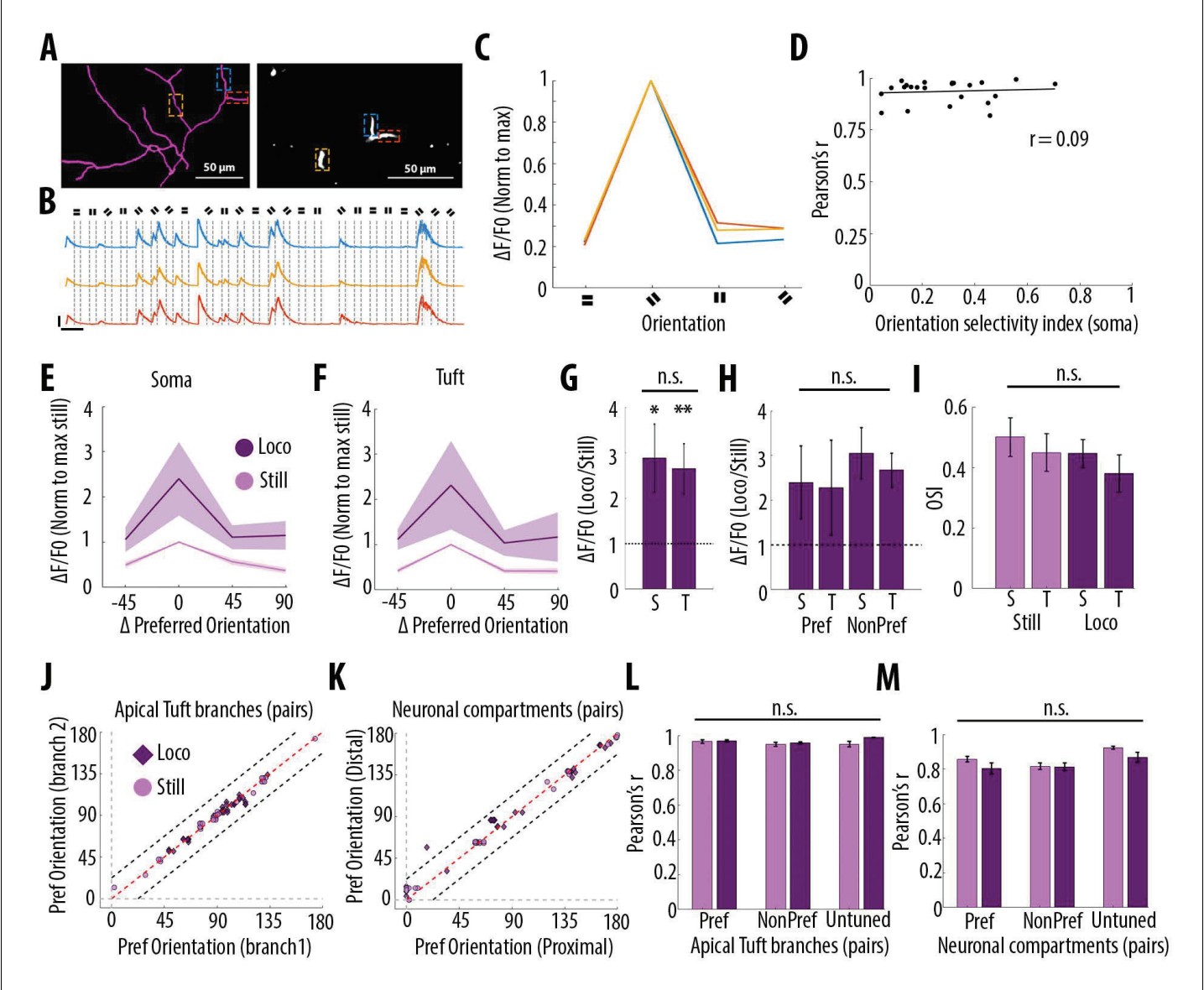

**Figure 6.** Orientation selectivity does not alter the relationship between somatic and dendritic calcium transients in layer 5 pyramidal neurons. (A) Morphological reconstruction (left) and two-photon image (right) of apical tuft branches of one individual neuron imaged during the presentation of drifting gratings. (B) Example GCaMP6s transients from the dendritic branches indicated by coloured dashed lines in panel A, during the presentation of drifting gratings of four different orientations. Dotted lines indicate the beginning and the end of drifting gratings stimulation. Scale bars 0.3 ΔF/F0 (normalised to max), 10 s. (C) Tuning curve showing calcium transients amplitudes (normalised to max) in response to the four grating orientations (average of 24 presentations of each grating, ), from the three branches indicated by coloured dashed lines in panel A. (D) Mean Pearson's correlation value of all imaged tuft dendritic branches per neuron, as a function of the orientation selectivity of the corresponding soma (each dot corresponds to one neuron; $r^2 = 0.09$, p=0.67, n = 23 neurons). The straight black line represents the best fit line (least square). (E) Tuning curve of somatic calcium signals from orientation selective neurons (somatic OSI >0.3; n = 15 neurons) during stationary (still, pink) and locomotion (loco, purple) periods. Responses are normalised to the preferred orientation during stationary periods. Thick line and shaded area represent the mean and SEM, respectively. (F) Same as E, for the corresponding apical tuft branches of the same neurons. For each neuron, responses from all imaged apical tuft branches were averaged (n = 68 apical tuft branches from 15 neurons). (G) Relative increase in calcium transient amplitudes during locomotion compared to stationary periods both in soma (S) and in corresponding apical tuft dendrites (T), during drifting grating presentations. Locomotion significantly increased responses to drifting gratings both in soma and corresponding apical tuft dendrites (Repeated Measures Two-way ANOVA on log transformed data, p=0.02 and p=0.008 for the effect of locomotion on soma and tuft responses, respectively, n = 15 neurons) without a significant difference between compartments (Paired t-test, p=0.38; mean (normalised to stationary)=2.89 and 2.65, sem = 0.74 and 0.56; n = 15 soma and 15 apical tufts including 68 branches). (H) Same as G for responses to the preferred (Pref) and non-preferred (NonPref) orientations (Repeated measures Two-way ANOVA on log-transformed data, p=0.07, 0.71 and 0.28 for orientation, compartment and interaction effects respectively; mean = 2.4 and 2.28, sem = 0.81 and 1.06 for

*Figure 6 continued on next page*

*Figure 6 continued*

soma and apical tuft at the preferred orientation; mean = 3.05 and 2.67, sem = 0.57 and 0.38 for soma and apical tuft at the non-preferred orientation; n = 15 soma and 15 apical tufts). (I) Orientation selectivity index was not significantly different between somatic and corresponding tuft dendrites responses, both during stationary (still) and locomotion (loco) periods (Repeated measures Two-way ANOVA, p=0.49, p=0.42 and p=0.45, for the effects of locomotion, neuronal compartment (soma vs tuft) and interaction, respectively; mean = 0.5; 0.45, sem = 0.06, 0.06 for soma and apical tuft during stationary and mean = 0.45; 0.38, sem = 0.05, 0.06 for soma and apical tuft during locomotion, respectively; n = 15 soma and 15 apical tuft including 68 branches). (J) Scatter plot of the preferred orientation in pairs of apical tuft branches belonging to the same neuron and imaged simultaneously, during stationary (pink circles) and locomotion (purple diamonds) periods (n = 31 pairs). Dashed red line: identity line; Dashed black lines: identity +/- 22.5 degrees. (K) Same as J for pairs of neuronal compartments imaged simultaneously. n = 33 pairs. (L) Pearson's correlation of calcium transients between pairs of apical tuft branches belonging to the same orientation-selective neuron, during the presentation of the preferred (Pref) and non-preferred (NonPref) grating orientations, both during stationary (pink) and locomotion (purple) periods (n = 31 pairs). Correlation values from branches belonging to non-selective neurons (OSI <0.3, untuned), during the presentation of all oriented gratings, are shown on the right columns (n = 10 pairs). Neither gratings orientation nor locomotion significantly affected the correlation between pairs of branches (Two-way ANOVA, p=0.20, p=0.07 and 0.38 for orientation, locomotion and interaction effect, respectively). (M) Same as L, for different neuronal compartments (Two-way ANOVA, p=0.27, p=0.09 and 0.43 for orientation, locomotion and interaction effect, respectively; n = 33 pairs for Pref, NonPref and 36 pairs for untuned). Error bars: SEM.

## Limitations of the use of calcium imaging to assess dendritic activity in awake behaving mice

Investigations of local dendritic activity in awake behaving mice have so far mainly relied on the use of genetically-encoded calcium indicators (*Xu et al., 2012*; *Cichon and Gan, 2015*; *Manita et al., 2015*; *Takahashi et al., 2016*; *Sheffield et al., 2017*; *Ranganathan et al., 2018*; *Kerlin et al., 2019*; *Sheffield and Dombeck, 2019*). The analysis of our data set showed that several experimental constraints may bias results towards signals mistakenly interpreted as local dendritic activity. Dense labelling was already demonstrated to strongly affect correlation values (*Xu et al., 2012*); even in cases of relatively sparse labelling of individual neurons, contamination from axons or dendrites of other labelled neurons may lead to local signals detected in single dendrites. In our data set, we had a sparse labelling and we systematically reconstructed our imaged neurons and excluded regions of interest where overlapping dendrites from other neurons were observed. Movement artefacts may also lead to inaccurate detection of local dendrite signals; these artefacts are more likely to be found during specific behaviour, such as locomotion, and therefore can bias the interpretation of the results. In our data set we used the activity-independent marker tdTomato to correct for motion artefacts and excluded the recordings in which tdTomato marker was not detected. For most of our imaged neurons, baseline fluorescence of both GCaMP6s and GCaMP6f could also be used for the same purpose. Another parameter that may cause artefactual local dendritic signals, is the length of the imaged dendritic segments (and therefore the size of the regions of interest used to extract the changes of fluorescence over time). Small regions of interest could include spines not clearly distinguishable from the corresponding dendritic shaft, especially when the spine is located above the dendrite at the imaged focal plane. As a consequence, in small regions of interests, spine signals may be incorrectly interpreted as local dendritic signals.

Differences in calcium buffering properties of individual compartments also create detection biases as they lead to different decay times of fluorescent calcium indicators between somatic and dendritic compartments (*Beaulieu-Laroche et al., 2019*; *Kerlin et al., 2019*). In order to account for this difference, we based our analysis on the detection of peak amplitudes of individual calcium transients. This detection depends on arbitrary thresholds; we thus showed the robustness of our results across different detection thresholds (*Figure 1—figure supplement 2*). An additional source of bias is linked to the affinities of calcium indicators. Previous studies have shown that dendritic calcium spikes can reliably be detected by calcium indicators with lower sensitivity than GCaMP6f and GCaMP6s (*Helmchen et al., 1999*; *Xu et al., 2012*). However, while it was shown that GCaMP6s and GCaMP6f could detect individual somatic action potentials in layer 2/3 neurons in vivo (*Chen et al., 2013*), this is not the case in layer 5 neurons neither for GCaMP6f (*Beaulieu-Laroche et al., 2019*), nor for GCaMP6s based on our own ex vivo experiments. We did however find that GCaMP6s was superior in detecting somatic calcium events during low frequency spiking (<25 Hz) (*Figure 4*). Nonetheless, the detection of somatic action potentials in layer 5 neurons in

vivo with GCaMP sensors may be biased towards larger events, and as a consequence, the attenuation from soma to tuft may be underestimated.

The relatively low temporal resolution of genetically-encoded calcium signals presents another key limitation to the study of dendritic calcium signals in vivo. In our study, we found branch-specific events to be rare. Whereas ex vivo studies have demonstrated that branch-specific stimulation can trigger localized $Ca^{2+}$ events detected by calcium indicators (*e.g.* *Cai et al., 2004*; *Sandler et al., 2016*; review: *Major et al., 2013*), the detection of these events in vivo is more challenging, principally because they may be masked by ongoing global dendritic events (*Kerlin et al., 2019*). Given the relatively low temporal resolution of GCaMP6s, dissociating global dendritic calcium signals from potential underlying branch-specific events is challenging (*Kerlin et al., 2019*). The contribution of branch-specific calcium signalling to ongoing neuronal activity in vivo therefore requires further investigation.

## Widespread, correlated calcium transients in apical tuft dendritic branches of layer 5 neurons

Our results show that in the apical tuft, calcium transients were highly correlated regardless of how distal the imaged branches were from the nexus. Branch-specific calcium transients were rare and limited to transients of the smallest amplitudes. Our results indicate that active dendritic integration associated with global calcium events in the apical tuft is widely recruited across different behavioural and sensory conditions. This widespread calcium activity throughout the layer 5 apical tuft may strongly influence plasticity and learning (*Magee, 1997*; *Golding et al., 2002*; *Kampa et al., 2006*; *Sjöström and Häusser, 2006*; *Remy and Spruston, 2007*; *Bittner et al., 2017*; *Bono and Clopath, 2017*; *Guerguiev et al., 2017*).

## High but asymmetric coupling of calcium signals from soma to apical tuft

Our results show an asymmetry in the coupling between somatic and apical dendritic calcium signals. Previous studies have shown that dendritic signals attenuate in a distance-dependent manner, suggesting that at least some of the calcium events generated in one compartment would attenuate as they reach the next compartment (*Stuart et al., 1997*; *Svoboda et al., 1997*; *Helmchen et al., 1999*; *Vetter et al., 2001*; *Larkum et al., 2009*; *Waters et al., 2003*). To estimate the lower bound proportion of calcium events generated in the tuft and in the soma, we investigated the proportion of attenuated events in both somatofugal and somatopetal directions. Our results show that while around 40% of somatic calcium transients decay by the time they reach the apical tuft, only 1.4% of apical tuft calcium events were not associated with somatic events (*Figure 5A*). These results indicate that at least 40% of somatic calcium transients were not triggered by apical tuft calcium events, and that these transients attenuated from the soma to the apical tuft. Based on our ex vivo recordings, we suggest that attenuated calcium transients are associated with low-frequency back-propagating somatic action potentials, below the critical frequency required to trigger dendritic electrogenesis (*Larkum et al., 1999a*; *Shai et al., 2015*). Attenuation may also involve active inhibitory mechanisms along the somato-dendritic axis (*Larkum et al., 1999a*; *Naka and Adesnik, 2016*).

A recent study reported a high and symmetric somato-dendritic coupling in V1 layer 5 neurons concluding that GCaMP6f events occurred concurrently in soma and dendritic compartments (*Beaulieu-Laroche et al., 2019*). The authors reported that 83.9% (median) of dendritic calcium signal (rise events) were paired with events in the soma while 73.4% of somatic events were paired with dendritic ones. In our study, we found a stronger asymmetry with 98.6% of dendritic events paired with somatic ones and 60 % of somatic events paired with dendritic ones. This discrepancy may be explained by differences in imaging conditions including a denser labelling, which increases the amount of background signals, and the use of GCaMP6f, which has a lower signal amplitude and signal-to-noise ratio than the GCaMP6s indicator used in our study, leading to a lower probability of detecting small events in layer 5 somata, as shown by our ex vivo recordings (*Figure 4G-H*).

In our study, we estimated that 60% of somatic calcium signals were highly correlated with global apical tuft calcium signals. Based on our ex vivo experiments, high somato-dendritic coupling would occur during either 1) high frequency back propagating action potentials or 2) strong apical tuft activity capable of driving somatic spiking. Coupling could in principle also be observed with lower

frequency back propagating action potentials, provided they are paired with tuft inputs (*Larkum, 2013*; *Manita et al., 2015*). Further experiments are needed to resolve these potential mechanisms in vivo, for example using voltage-sensitive dyes (*Villette et al., 2019*; *Roome and Kuhn, 2018*; *Adam et al., 2019*) or dendritic electrophysiological recordings (*Moore et al., 2017*), that would provide the temporal resolution to resolve the different types of dendritic events.

Notably, both apical tuft signals and the somato-dendritic coupling may differ between different subtypes of layer 5 pyramidal neurons. It is known that at least two main types exist: intratelencephalic neurons which connect cortical areas, and pyramidal tract neurons which project to multiple subcortical areas (*Harris and Shepherd, 2015*; *Gerfen et al., 2018*). These two types display different morphologies (*Groh et al., 2010*) and receive different types of inputs (*Young, 2019*). The neurons included in this study had their soma located at various depths within layer 5 (median 528 μm, see *Figure 3—figure supplement 1*) and mainly displayed thick-tufted morphology, characteristic of pyramidal tract neurons. Since we selected the imaged neurons visually based on their GCaMP6 fluorescent signal, our sampled is likely biased towards layer 5 neurons with thick trunk and thick-tufted morphology.

## Integration of visual and locomotion-related inputs in layer 5 neurons in awake behaving mice

Our results show that, in our passive viewing conditions, changes in visual inputs (darkness versus drifting gratings) and locomotion-related inputs do not affect the relationship between somatic, trunk and apical tuft calcium signals nor the prevalence of branch-specific dendritic events in the apical tuft. These results are in agreement with a previous study showing that somato-dendritic correlations of calcium transients remain unchanged by visual stimulation and locomotion (*Beaulieu-Laroche et al., 2019*). It is however possible that in both studies, untracked behavioural states (e.g. arousal, attention) during stationary and locomotion periods may differentially shape somato-dendritic activity.

It is known that the prevalence and the dynamics of synaptic inputs received by layer 5 pyramidal neurons strongly vary between passive sensory stimulation and active learning tasks (*Xu et al., 2012*; *Hong et al., 2018*). Apical tuft dendrites of layer 5 neurons receive a barrage of thousands of synaptic inputs in vivo (*Stuart and Spruston, 2015*) and it was shown that V1 neurons receive synaptic inputs conveying information not only about visual stimuli but also about non-visual variables (e. g. arousal/attention, motor activity, reward-related and spatial information) (*Pakan et al., 2016*; *Pakan et al., 2018a*; *Pakan et al., 2018b*). Finally, in addition to excitatory inputs, both inhibitory and neuromodulatory inputs were shown to modulate electrical interactions between soma and dendrites in layer 5 neurons (*Larkum et al., 1999a*; *Silberberg and Markram, 2007*; *Labarrera et al., 2018*; *Williams and Fletcher, 2019*). It is therefore possible that during the active learning of a behavioural task, the synaptic inputs associated with this learning process would lead to different mechanisms of dendritic integration than during passive viewing. For example, dendritic integration of multiplicative combinations of sensory and motor inputs in the tuft dendrites of layer 5 pyramidal neurons has been described during an active sensing task in mouse somatosensory cortex (*Ranganathan et al., 2018*). In addition, it was proposed that changes in neuronal representations during learning would rely on active dendritic signals generating calcium plateau potentials (*Bittner et al., 2015*; *Grienberger et al., 2017*). In that case, somato-dendritic coupling may change during the course of learning (*Sheffield et al., 2017*). This remains to be tested in the visual cortex. Similarly, somato-dendritic coupling may evolve during postnatal development when sensori-motor associations are formed. Recently, it was shown in the mouse primary visual cortex, that dendrites of layer 2/3 neurons increase their coupling during adulthood, as a consequence of decreased responsiveness of dendrite-targeting interneurons to locomotion-related inputs (*Yaeger et al., 2019*). Further investigations are needed to reveal these mechanisms in layer 5 visual cortical neurons.

## Materials and methods

### Animals

All experiments and procedures involving animals were approved by the University of Edinburgh Animal Welfare and the ethical review board (AWERB). Experiments were performed under the

appropriate personal and project licenses from the UK Home Office in accordance with the Animal (Scientific Procedures) act 1986 and the European Directive 86/609/EEC on the protection of animals used for experimental purposes. Adult male and female mice, aged between 8 to 10 weeks, were obtained from Jackson Laboratory, ME, USA (B6.Cg-Gt(ROSA)26Sor$^{tm14(CAG-tdTomato)Hze}$/J [RRID: IMSR_JAX:007914]). Mice were group caged in groups of 2–6 animals, with a running wheel and on a reverse 12:12 hr light/dark cycle, with ad libitum access to food and water.

## Surgical procedures

### Viral delivery of GCaMP6

To obtain sparse labelling of excitatory neurons we used a Cre-dependent approach, by co-injecting viral constructs AAV1.CamKII 0.4.Cre.SV40 (Penn Vector core catalogue No. 105558-AAV1) diluted at 1:10000 or 1:20000 with either AAV1.Syn.Flex.GCaMP6f.WPRE.SV40 (Penn Vector Core, catalogue No. 100833-AAV1) or AAV1.Syn.Flex.GCaMP6s.WPRE.SV40 (Penn Vector Core, catalogue No. 100845-AAV1) diluted 1:10 in the final solution. All dilutions were made in sterile artificial cerebrospinal fluid (ACSF). AAV1.Syn.Flex.GCaMP6f.WPRE.SV40 was used for single plane imaging of the apical tuft while AAV1.Syn.Flex.GCaMP6s.WPRE.SV40 was used for semi-simultaneous two-planes imaging. Animals were initially anesthetized with 4% Isoflurane and subsequently maintained on 1–2% isoflurane throughout the procedure. Body temperature was maintained at physiological levels using a closed-loop heating pad. Eye cream (Bepanthen, Bayer) was applied to protect the eyes from dryness and light exposure.

After induction of anaesthesia, mice were shaved and mounted onto a stereotaxic frame (David Kopf instruments, CA, USA). An analgesic was administered subcutaneously (Vetergesic, buprenorphine, 0.1 mg/kg of body weight). Viral injections were performed in the primary visual cortex of left hemispheres (centred around 2.5 mm lateral from midline and 0.5 mm anterior to lambda) at 650 and 500 μm from brain surface using a glass pipette (20 μm tip diameter, Nanoject, Drummond Scientific), coupled to a Nanoject II (Drummond Scientific). A total volume of 55.2 nl was injected across the two depths (6 × 4.6 nl at each location with 30 s intervals between each injection to allow sufficient time for diffusion). After each injection, pipettes were left in situ for an additional 5 min to prevent backflow. The scalp was then sutured (Ethicon, Ethilon polyamide size 6) and the animal monitored during recovery in a heated cage before returning to its home cage for 2–3 weeks.

### Headplate and imaging window

Under anaesthesia (isoflurane), mice were shaved and mounted onto a stereotaxic frame (David Kopf instruments, CA, USA). Analgesic and anti-inflammatory drugs were administered subcutaneously (Vetergesic, buprenorphine, 0.1 mg/kg of body weight; Carpaphen, Carprieve, 5 mg/kg of body weight; Dexamethasone, Rapidexon, 2 mg/kg of body weight). For a cranial window over V1, a section of scalp was removed, the underlying bone was cleaned before a rectangular craniotomy of about 2 × 1.5 mm was made over the left primary visual cortex (centred around 2.5 mm lateral and 0.5 mm anterior to lambda). The dura was kept intact. The craniotomy was then sealed with a glass cover slip and fixed with cyanoacrylate glue. To stabilize the brain during imaging, we used a triple glass window by stacking three cover-slip glasses (Menzel-Glaser 24 × 32 mm # 0) on top of each other. These were glued together using an optically clear UV-cured glue (Norland Optical adhesive). Overall, the thickness of the glass window inserted through the cranial window, was comparable to the thickness of the skull at the time of imaging. A custom-built metal headplate was fixed on top of the skull with glue and cemented with dental acrylic (Paladur, Heraeus Kulzer). At the end of the procedure, a single dose of 25 mL/kg of Ringer's solution was injected subcutaneously to rehydrate the animal after the procedure. The animal was then released from the head fixation and returned to a heated recovery cage, until full motor capacity was recovered.

## In vivo two-photon imaging data acquisition

Imaging was performed using a 25x Objective (Olympus). The excitation wavelength of the laser was set to 920 nm. Layer 5 pyramidal neurons were imaged between 500 and 650 μm below the brain surface and followed up to their distal tuft dendrites along the apical trunk. Neurons which had their nucleus filled with GCaMP6 or had blebbed dendrites were excluded. Imaging was performed using a 570 nm short-pass dichroic mirror and two single-band pass filters, a 525/50 and a 620/60

(Scientifica). At the end of each imaging session, a z-stack was acquired to allow offline morphological reconstruction of the imaged neuron. Reconstructions were done using the ImageJ plugin Simple Neurite Tracer.

Habituation and imaging started 2–3 weeks after AAV injection. Mice were habituated to head-fixation in the dark for 45 min and began to run freely on a polystyrene cylindrical treadmill (20 cm diameter, on a ball-bearing mounted axis). Running speed on the treadmill was continuously monitored using an optical encoder (E7P, 250cpr, Pewatron, Switzerland) that was connected to a data acquisition device (National Instrument, UK). Data were recorded with custom-written software in LabView (National Instrument, UK) and analysed in MATLAB (Mathworks, MA).

Single-plane two-photon imaging data were acquired using a custom-built galvo-resonant scanning system with a Ti:Sapphire pulsing laser (Chameleon Vision-S, Coherent, CA, USA;<70 fs pulse width, 80 MHz repetition rate) at 120 Hz, with a custom-programmed LabVIEW-based software (version 8.2; National Instruments, UK). Multi-plane data were acquired using an ultra-fast, solid-state, single 100 fs pulse width laser (InSight DeepSee, SpectraPhysiscs, CA, USA) and a FemtoSmart Dual two-photon microscope (Femtonics, Budapest, Hungary). Two focal planes (512 × 165 pixels), with an average distance of 170 μm in Z, were imaged at a frequency of 96 frames/s (48 frames/s per plane) using a Piezo objective positioner kit (P725.4CA, Physic Instruments, Germany), switching between planes at 9.6 Hz (4.8 Hz per plane).

## Visual stimulation

Visual stimuli were generated using the Psychophysics Toolbox package (*Brainard, 1997*) for MATLAB (Mathworks, MA) and displayed on a custom-modified, backlit LED monitor (51 × 29 cm, Dell, UK), which was placed 20 cm from the right eye, covering 104°×72° of the visual field. Visual stimulation trials consisted of drifting full-field square-wave gratings for 3 s (spatial frequency of 0.05 cycles per degree, 1.5 Hz, eight equally spaced directions in randomized order, contrast 80%, mean luminance 37 cd/m$^2$). Drifting grating presentations were separated by 4 s periods of isoluminant grey stimuli. We acquired 8–12 trials in darkness and 12–20 trials during visual stimulation for each imaged field of view. Time stamps for the onset of every stimulus were recorded and aligned to imaging frames using custom-built Matlab scripts.

## In vivo data analysis

### Image analysis and signal extraction

To correct for brain motion after image acquisition, we used 2D plane translation-based image alignment (SIMA 1.2.0, sequential image analysis; *Kaifosh et al., 2014*). We 3D reconstructed each imaged neuron and defined regions of interest (ROIs) corresponding to neuronal cell body and dendritic segments manually. Due to sparse labelling, signal contamination was negligible in most cases. ROIs that were contaminated by labelled structures from other neurons were excluded from further analysis.

The pixel intensity within each ROI was averaged to create a raw fluorescence time series F(t). Baseline fluorescence F0 was computed for each neuron by taking the fifth percentile of the smoothed F(t) (1 Hz lowpass, zero-phase, 60th-order FIR filter) over each trial (F0(t)), averaged across all trials. As a consequence, the same baseline F0 was used for computing the changes in fluorescence in darkness and during visual stimulation. The change in fluorescence relative to baseline, ΔF/F0 was computed by taking the difference between F and F0(t) and dividing by F0.

Single plane data was acquired at 120 Hz, and subsequently downsampled to 5 Hz for signal processing. Multi-plane data were analysed at 4.8 Hz.

### Calcium transient analysis

We first estimated noise levels by filtering our ΔF/F0 signal using a 9[th] order, zero-phase, high-pass filter at 0.6 Hz (Matlab function *filter*). We then estimated the standard deviation of the filtered signal and used a threshold of 2.8 of this standard deviation to detect individual calcium transients, using the built-in Matlab function *findpeaks*. We tested the robustness of our results to different thresholds +/- 30% of the selected threshold (*Figure 1—figure supplement 2*).

For each peak found in any branch or compartment, we defined a calcium transient as coincident in another branch or compartment when it occurred in a time window of 3 s around the frame where

the first peak was originally detected (2 s before, one after). As a consequence, peaks detected as coincident could not be further apart than 2 s. *Figure 1—figure supplement 2* shows the distribution of time intervals between events detected as coincident. The mean time interval was close to 0 ($2.19xe^{-4}$ seconds) with standard deviation of 0.28 s, meaning that 95% of all coincident events were found within a time interval of 0.56 s. Each calcium transient could only be considered coincident with one transient. The peak amplitude of a calcium transient was determined by taking the difference between the ΔF/F0 amplitude at the frame in which the peak was detected and a local minimum in a 2 s, backward sliding window. Correlation values were calculated as the Pearson's correlation values between peak calcium transient amplitudes in pairs of branches or compartments. Whenever a peak was detected in only one compartment and not in the other (compartment-specific events), the correlation was made between the peak amplitude in one compartment and the difference between the maximum and minimum ΔF/F0 values in a 3 s window centred around the frame of the detected peak in the other compartment. Shuffling values were obtained by randomly shuffling the order of the events in one branch or compartment of each pair. Residuals were extracted by calculating the robust linear regression line between normalised event amplitudes in each compartment, and then extracting the distance, along the y-axis, of each individual point from this robust line.

## Locomotion analysis

Changes in the position of the cylindrical treadmill (sampled at 12,000 Hz) were interpolated to match the rate of imaging. To define stationary and locomotion periods we used the following criteria: Stationary corresponded to periods where the instantaneous speed (as measured at the 40 Hz sampling rate) was less than 0.1 cm/s. Locomotion corresponded to periods meeting three criteria: instantaneous speed $\geq$0.1 cm/s, 0.25 Hz lowpass filtered speed $\geq$0.1 cm/s, and an average speed $\geq$0.1 cm/s over a 2 s window centered at this point in time (*Pakan et al., 2016*). Any interlocomotion interval shorter than 500 ms was also labelled as locomotion. Stationary periods less than 3 s after or 0.2 s before a period of locomotion were removed from the analysis. For *Figure 5—figure supplement 1*, behavioural transitions were defined as time windows including: (1) 2 s before the onset of locomotion as defined above and 1 s after the onset, as well as (2) 1 s before the offset of locomotion and 20 s after the offset (unless another locomotion period began before the 20 s) (*Vinck et al., 2015*). For this analysis, behavioural transitions were excluded from stationary and locomotion periods.

## Orientation selectivity

To determine the specific stimulus response parameters of each neuron to the oriented gratings, the ΔF/F0 during each presented oriented stimulus was first averaged across all trials. The preferred orientation of each neuron was the orientation that elicited the maximal response when averaged across all trials. The orientation selectivity index (OSI) was calculated as ($O_{pref}$-$O_{orth}$)/($O_{pref}$+$O_{orth}$) where $O_{pref}$ represents the mean ΔF/F0 value during the presentation of the preferred orientation across trials and $O_{orth}$ represents the mean ΔF/F0 value during the presentation of the orientation orthogonal to the preferred one. For the detailed analysis of orientation preference in *Figure 6J and K*, the preferred response angle was estimated by calculating the argument of the resultant vector (V) in the complex plane, which was given by:

$$V = \frac{\sum_k R(\theta_k) e^{2i\theta_k}}{\sum_k R(\theta_k)}$$

where $R(\theta_k)$ is the mean ΔF/F0 response to angle $\theta_k$ (*Mazurek et al., 2014*).

## Ex vivo patch-clamp recordings and calcium imaging

C57BL/6 mice (8–10 week old) were injected with either flexed GCaMP6s (AAV1.Syn.Flex.GCaMP6s. WPRE.SV40; dilution 1:10; Penn Vector Core, catalogue 384 No. 100845-AAV1) or flexed GCaMP6f (AAV1.Syn.Flex.GCaMP6f.WPRE.SV40; dilution 1:10; Penn Vector Core, 383 catalogue No. 100833-AAV1) along with CaMKII-Cre (AAV1.CamKII 0.4.Cre.SV40; diluted 1:100 or 1:1000; Penn Vector core catalogue No. 105558-AAV1) targeted to Layer 5 in V1 (see Viral Delivery of GCaMP6 in Materials and methods for details on the injection parameters). 3–4 weeks following injection,

animals were decapitated under isoflurane anaesthesia. The brain was extracted and coronally sliced into 400 µm thick sections in ice cold, carbogenated (95% $O_2$, 5% $CO_2$) dissection media (in mM: 87 NaCl, 75 sucrose, 2.5 KCl, 25 NaHCO3, 1.25 NaH2PO4, 25 glucose, 0.5 CaCl2, 7 MgCl2,) using a vibratome (VT1200s, Leica, Germany). Slices containing V1 were isolated and allowed to recover for 30–60 min in heated (35°C) carbogenated ACSF (in mM: 125 NaCl, 2.5 KCl, 25 NaHCO3, 1.25 NaH2PO4, 25 glucose, 2 CaCl2, 1 MgCl2) before being stored at room temperature.

During recordings, slices were perfused with heated (33°C), carbogenated ACSF. Layer 5 GCaMP6-positive cells were patched with borosilicate glass electrodes (4–7 MΩ) filled with K-gluconate internal solution (in mM: 130 K-Gluconate, 10 KCl, 2 MgCl2, 10 EGTA, 10 HEPES, 2 Na2-ATP, 0.3 Na2-GTP; pH 7.2–74, 290–300 mOsm), containing 0.1 mM Alexa 594 (ThermoFisher, UK) to facilitate visualization of cells. Cellular electrophysiology was recorded using a Multiclamp 700B amplifier and associated pCLAMP software (Molecular Devices). Data were acquired at 10 kHz and filtered at 3 kHz. For somatic stimulation, 10 current pulses (2–5 ms, 1–1.5nA) were delivered at 5, 10, 25, 50, 100 and 200 Hz. Each pulse was of sufficient amplitude to trigger an action potential. For dendritic stimulation, a monopolar tungsten electrode was placed in L1. Using a constant current stimulator (Digitimer, UK) 10 stimulation pulses (100 µs) were delivered at 5, 10, 25, 50, 100 and 200 Hz at a stimulation intensity 25, 50, 75% and 100% of spiking threshold in the soma. Suprathreshold stimulation was often accompanied by an enhanced afterdepolarization reminiscent of a dendritic spike (Larkum et al., 1999a; Shai et al., 2015). In sham stimulation trials, no stimulus was given.

Cells were imaged with a 25x Olympus Objective (1.05 NA; XLPlan N, Olympus) using a two-photon microscope (Femtonics) equipped with a Ti:sapphire pulsing laser (Chameleon Vision-S, Coherent, CA, USA) which was tuned at 920 nm for GCaMP6 imaging, and 800 nm for Alexa 594 imaging. During somatic and dendritic stimulation, xy scans of GCaMP6 signals at the soma and distal apical dendrite (nexus) were acquired at a rate >120 Hz. Each imaging trial was 6 s in duration, with stimulation commencing after a 1 s baseline recording.

Images were downsampled to 20 Hz and analysed using ImageJ software and custom MATLAB scripts. Electrophysiological traces were analysed using pCLAMP software and custom MATLAB scripts. $\Delta F/F0$ was calculated as: $(F – F0)/(F0 – F_{background})$ where F0 was the mean baseline fluorescent signal during the 1 s baseline recording, and $F_{background}$ was the background fluorescence. F0 in the distal dendrites was often very low ex vivo, leading to inflated $\Delta F/F0$ values. Consequently, $\Delta F/F0$ signals in the soma and dendrite were standardized to the maximum signal evoked in each compartment. This allowed for fairer comparisons between compartments, and across cells. Calcium transient amplitudes were measured as the peak $\Delta F/F0$ observed within 1 s after the cessation of stimulation. These values were then subtracted by the average peak $\Delta F/F0$ measured in sham stimulation trials. The procedure was repeated for each sham stimulation trial in order to derive a null distribution of peak amplitudes for statistical comparison. Calcium events were also classified as successfully detected if they exceeded 2.8 standard deviations from the expected null, similar to what was used to analyse our in vivo data. An event was considered compartment specific if it was detected in one compartment but not in the other.

## Statistics

All analysis was performed either using MATLAB (Mathworks, MA), or GraphPad Prism 8. All error bars in the figures represent standard error of the mean (SEM). Statistical tests and independent samples are described in figure legends.

## Acknowledgements

We thank the GENIE Program and the Janelia Research Campus, specifically V Jayaraman, R Kerr, D Kim, L Looger, and K Svoboda, for making GCaMP6 available. We would also like to thank Peter Kind, for allowing access to equipment and reagents required for ex vivo experiments. This work was funded by the Wellcome Trust and the Royal Society (Sir Henry Dale fellowship to NR), the Marie Curie Actions of the European Union's FP7 program (MC-CIG 631770 to NR), the Shirley Foundation, the Patrick Wild Center and the RS MacDonald Charitable Trust Seedcorn Grant (to NR), the Simons Initiative for the Developing Brain (to NR), the Royal Commission for the Exhibition 1851 (Research Fellowship to ZP) and the University of Edinburgh (PhD scholarship to VF).

## Additional information

### Funding

| Funder | Grant reference number | Author |
| --- | --- | --- |
| Wellcome | 102857/Z/13/Z | Nathalie L Rochefort |
| Royal Society | 102857/Z/13/Z | Nathalie L Rochefort |
| University Of Edinburgh | PhD fellowship | Valerio Francioni |
| Simons Initiative for the Developing Brain | Project grant | Nathalie L Rochefort |
| Seventh Framework Programme | CIG 631770 | Nathalie L Rochefort |
| RS MacDonald Charitable Trust | Seedcorn Grant | Nathalie L Rochefort |
| Royal Society | Royal Commission for the Exhibition 1851 | Zahid Padamsey |

The funders had no role in study design, data collection and interpretation, or the decision to submit the work for publication.

### Author contributions

Valerio Francioni, Data curation, Software, Formal analysis, Investigation, Visualization, Methodology; Zahid Padamsey, Formal analysis, Investigation, Methodology, Performed ex vivo experiments; Nathalie L Rochefort, Conceptualization, Resources, Data curation, Formal analysis, Supervision, Funding acquisition, Investigation, Methodology, Project administration

### Author ORCIDs

Nathalie L Rochefort (iD) https://orcid.org/0000-0002-3498-6221

### Ethics

Animal experimentation: All experiments and procedures involving animals were approved by the University of Edinburgh Animal Welfare and the ethical review board (AWERB) and performed under the appropriate PIL and PPL license from the UK Home Office in accordance with the Animal (Scientific Procedures) act 1986 and the European Directive 86/609/EEC on the protection of animals used for experimental purposes.

### Decision letter and Author response

Decision letter https://doi.org/10.7554/eLife.49145.sa1
Author response https://doi.org/10.7554/eLife.49145.sa2

## Additional files

### Supplementary files

• Transparent reporting form

### Data availability

Raw data (changes of fluorescence over time) are provided in all the main figures (figure 1-4) and in two supplementary figures. Additionally, we provide two data source videos (Video 1 and Video 2). Due to the large volume of imaging data sets, all raw data (videos) are on a dedicated server from Rochefort lab and are available upon request. The Center for Discovery Brain Sciences, University of Edinburgh, is setting up a repository for published data sets that will be used for the data included in this manuscript, as soon as it is available. All analyses were performed using custom-written scripts in MATLAB, which are freely available via GitHub repository (https://github.com/rochefort-lab/

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
