## [Decision Letter]

**Acceptance summary:**

Francioni et al. have investigated the coupling of calcium transients across the soma and the dendritic tree of layer 5 pyramidal neurons by recording from the primary visual cortex of awake mice. They show that (1) calcium transients are highly correlated across different dendritic compartments and the soma, and (2) the somato-dendritic coupling is asymmetric in that ~40% of somatic events are attenuated as function of distance towards the distal tuft but not in the opposite direction. Furthermore, the two findings appear to be independent of the locomotive state of the animal or the presentation of visual stimuli. While the first main finding of the correlated somato-dendritic signals that occur irrespectively of visual stimulation and locomotion, recapitulates a recent study (Beaulieu-Laroche et al., 2019), the second key finding further extends the prior observations in demonstrating that a large fraction of somatic and/or proximal dendritic events are uncoupled from the global dendritic events. Overall this is a carefully executed study focusing on an important problem in dendritic processing in awake behaving mice.

**Decision letter after peer review:**

Thank you for submitting your article "High and asymmetric somato-dendritic coupling of V1 layer 5 neurons independent of visual stimulation and locomotion" for consideration by *eLife*. Your article has been reviewed by three peer reviewers, one of whom is a member of our Board of Reviewing Editors, and the evaluation has been overseen by Joshua Gold as the Senior Editor. The reviewers have opted to remain anonymous.

The reviewers have discussed the reviews with one another and the Reviewing Editor has drafted this decision to help you prepare a revised submission.

Francioni and Rochefort have investigated the coupling of calcium transients across the soma and the dendritic tree of layer 5 pyramidal neurons by recording from V1 of awake mice. The two major findings are: (1) calcium transients are highly correlated across different dendritic compartments and the soma, and (2) the somato-dendritic coupling is asymmetric in that ~40% of somatic events are attenuated as function of distance towards the distal tuft but not in the opposite direction. Furthermore, the two findings appear to be independent of the locomotive state of the animal or the presentation of visual stimuli.

The first of the two main findings – of correlated somato-dendritic signals that occur irrespectively of visual stimulation and locomotion – recapitulates the very recent study by Beaulieu-Laroche et al. (Neuron 2019). The second of the two findings further extends the observations by showing that a large fraction of somatic/proximal dendritic events are uncoupled from the global dendritic events, thereby suggesting that the generation of global dendritic events require the integration of different source of input. Overall this is a carefully executed study focusing on an important problem in dendritic processing in awake behaving animal.

However, the crucial concern raised by all three reviewers relates to the idea of the apical tuft acting as a single computational unit. Such a conclusion is not compellingly supported by the results shown. If the calcium imaging only picks up large calcium spikes (cf. Beaulieu-Laroche et al., 2019), then it remains possible that single dendritic branches integrate synaptic input non-linearly via local electrical events below the imaging detection limit, which then lead to a large nexus spike that can be imaged. In this case, the apical tuft could still contain multiple computational units, and the authors cannot exclude this possibility with the current data. In fact, the data show that around 15% of small calcium transients are branch specific, which could correspond to local spikes that did not cause a nexus spike. Therefore, the strength of somato-dendritic coupling is likely to be over-estimated in this study. Moreover, there could also be other possibilities for not observing local branch signals. During awake behaving conditions, untracked behavioral events or cognitive processes may produce dendritic spikes, and therefore it is not clear what behaviorally-relevant event is triggering the calcium transients.

In order to address the crucial concerns raised above, it is suggested that the authors (1) tone down the interpretations of the data and draw conclusions that are directly supported by the observations under the specific experimental conditions and (2) further analyze the relation between dendritic calcium signals and behavioral events. What kind of behavioral events are responsible for the somatic signals that failed to invade distal dendrites? Also, with respect to the distal dendritic calcium signals that always correlate to the proximal/somatic compartment events, the authors should modify their view and focus on the relationship between the global dendritic signals and behavioral events. It would be valuable if the authors could analyze how the interaction between locomotion and visual stimulation influences the dendritic global signals, and this may provide new insights into dendritic computation. Additional major comments and requests are listed below.

Provided the authors are able to fully address the major concerns, the manuscript should be of sufficient interest to *eLife*. Assuming that the authors already have the data concerning the dendritic calcium signals and behavioral events needed for the analysis, it is expected that 2 months should be sufficient to complete the revision.

Essential Revisions:

1) The observation that behavior or visual stimulation did not influence the coupling between distal tuft and soma may seem intriguing. But the current behavior paradigm does not fully engage the animal in a well-controlled task. There could be untracked behavioral events other than locomotion, and there could be covert cognitive processes going on even in the darkness without running. These untracked behavioral events or cognitive processes may produce global dendritic spikes. Even without locomotion and visual stimulation, there can still be hidden variables that would produce local dendritic signals that cannot be isolated. The authors should provide a more comprehensive and cautious interpretation to their observations.

2) The analysis of dendritic calcium signals is relatively crude, regarding their relationship to behavioral parameters. The two most important variables in the behavioral paradigm are the visual stimulation and locomotion. When studying dendritic integration, an obvious key question is how dendrites in the large L5 neurons integrate these two inputs. How might locomotion influence visual responses in the dendrites? Would there be a gain modulation of visual responses by locomotion, and if so, could such gain modulation be mediated by dendritic integration? The authors should at least quantify the relationship between dendritic signals and locomotion, and compare dendritic responses to the same visual stimuli between stationary and locomotive state. The authors showed orientation tuning of dendritic responses and its relationship to correlations, which is nice. But how would locomotion influence such orientation tuning and correlations? Currently there is only Figure 1H showing the mean fluorescence changes under different conditions, which is not very informative.

3) It's not clear what the biophysical interpretation of the asymmetric coupling is. Electrically, in a passive dendritic tree, proximal to distal propagation is more efficient than distal to proximal propagation. Here the data show an asymmetry of calcium transients in the reverse direction. In principle this could be due to distal calcium transients being associated with large dendritic spikes, which would always trigger somatic action potentials and thus an apparent strong "coupling", while proximal transients might include action potentials that do not backpropagate fully into the apical dendrite. However, looking at the data in Figure 3, there are examples of proximal/distal transient pairs that are smaller than large proximal transients that don't show up in the apical dendrites. What is the authors interpretation? Is the asymmetry coming from inhibition?

4) The authors should provide some measure of the relationship between calcium and voltage by performing whole-cell recordings from soma and dendrites of GCamp6-expressing L5 cells in vitro, as Beaulieu-Laroche et al. have done. This is particularly important because the asymmetry in the somato-dendritic coupling reported here contradicts the findings of Beaulieu-Laroche et al. While the difference could be due to the indicator, it would be important to experimentally clarify where the difference comes from.

5) The conclusion about the correlation between distal dendrites and soma is based on indirect measurements. In Figure 2, the authors showed the correlations between compartments of neighboring focal planes. But presumably due to the technical limitation, there is no direct measurement of the correlation between the distal tuft branches and somatic signals. The author should at least provide certain estimation about to what extent the multiple correlations between neighboring compartments may tell about the direct correlation between distal branches and soma.

6) Although the authors used CaMKII to restrict the expression of Cre in excitatory neurons, this study did not distinguish different subtypes of L5 pyramidal neurons. It is known that there are at least two or more different subtypes of L5 neurons with distinct dendritic morphologies, dendritic integrative properties, and downstream projection patterns. For example, the cortico-cortical projection L5 neurons are mostly thin tufted neurons, showing less bursting activity, which may be attributable to differences in the properties of dendritic spikes. Additionally, the distal tuft of the thick tufted neurons may be more compartmentalized than the thin tufted neurons, and show large dendritic calcium spikes. How such differences in different subtypes of L5 neurons may influence the observations in the current study? At least, the authors should include a discussion on this issue.

[Editors' note: further revisions were requested prior to acceptance, as described below.]

Thank you for resubmitting your work entitled "High and asymmetric somato-dendritic coupling of V1 layer 5 neurons independent of visual stimulation and locomotion" for further consideration by *eLife*. Your revised article has been reviewed by two of the original reviewers and overseen by Joshua Gold (Senior Editor) and a Reviewing Editor.

The revised manuscript has been significantly improved with inclusion of additional experiments and analysis and careful editing of the text. However, there is a remaining issue that needs to be addressed prior to further consideration of the manuscript. This requires an additional set of ex vivo experiments that is crucial to the main conclusions of the paper, which we think are feasible to complete within a month.

The reviewers have pointed out the shortcomings related to the new ex vivo experiments (Figure 4). Specifically, the use of suprathreshold dendritic stimulation that also evokes somatic APs strongly activates both dendrites and soma, and produces an apparent strong coupling in the dendrite to soma direction. However, given that dendritic stimulation can evoke somatic APs, bpAPs could have interacted with local dendritic activity. Therefore, it is not clear if in the present imaging conditions, how local dendritic events lead to global dendritic spikes and if there are dendritic events that do not cause the soma to fire. In order to support the claim of the rarity of such events in vivo, or to test for the possibility that they cannot be detected, the authors need to show by imaging using a range of subthreshold dendritic stimulations given to layer 1 in the ex vivo preparation, the dendrite to soma coupling as a function of the level of dendritic calcium activity.

---

## [Author Response]

[…] However, the crucial concern raised by all three reviewers relates to the idea of the apical tuft acting as a single computational unit. Such a conclusion is not compellingly supported by the results shown. If the calcium imaging only picks up large calcium spikes (cf. Beaulieu-Laroche et al., 2019), then it remains possible that single dendritic branches integrate synaptic input non-linearly via local electrical events below the imaging detection limit, which then lead to a large nexus spike that can be imaged. In this case, the apical tuft could still contain multiple computational units, and the authors cannot exclude this possibility with the current data. In fact, the data show that around 15% of small calcium transients are branch specific, which could correspond to local spikes that did not cause a nexus spike. Therefore, the strength of somato-dendritic coupling is likely to be over-estimated in this study.

We have removed all instances suggesting that the apical tuft acts as a single computational unit. We fully agree with the limitations of calcium imaging and provide a detailed description of these limitations in the Discussion (subsection “Limitations of the use of calcium imaging to assess dendritic activity in awake behaving mice”).

We have performed additional analysis to quantify which proportion of the tuft events detected as branch-specific would also be detected in the apical trunk. On average, we found that 60% of the events detected as branch-specific were also found in the trunk (Figure 2—figure supplement 2), reducing even further the proportion of events exclusively detected in a single dendritic branch. We have edited the text to include these results (subsection “Calcium signals are highly correlated throughout all compartments of individual layer 5 neurons”).

Moreover, there could also be other possibilities for not observing local branch signals. During awake behaving conditions, untracked behavioral events or cognitive processes may produce dendritic spikes, and therefore it is not clear what behaviorally-relevant event is triggering the calcium transients.In order to address the crucial concerns raised above, it is suggested that the authors (1) tone down the interpretations of the data and draw conclusions that are directly supported by the observations under the specific experimental conditions.

We fully agree that untracked behavioural events or cognitive processes may affect somato-dendritic coupling. We have extended our Discussion to emphasize that while we did not find an impact of visual stimulation or locomotion on somato-dendritic coupling, it is possible that untracked behavioural states (e.g. arousal, attention) may differentially shape somato-dendritic calcium signals (subsection “Integration of visual and locomotion-related inputs in layer 5 neurons in awake behaving mice”). We also emphasize that the prevalence and the dynamics of dendritic calcium signals and somato-dendritic coupling may strongly vary between passive sensory stimulation and active learning.

2) Further analyze the relation between dendritic calcium signals and behavioral events. What kind of behavioral events are responsible for the somatic signals that failed to invade distal dendrites? Also, with respect to the distal dendritic calcium signals that always correlate to the proximal/somatic compartment events, the authors should modify their view and focus on the relationship between the global dendritic signals and behavioral events. It would be valuable if the authors could analyze how the interaction between locomotion and visual stimulation influences the dendritic global signals, and this may provide new insights into dendritic computation.

We have performed further analyses of our data set on the impact of behavioural state transitions (between stationary and locomotion) and of drifting gratings orientation on apical tuft dendrites and somatic responses. These results are shown in the newly added Figure 5B, Figure 5—figure supplement 1 and 2 and Figure 6.

Our results show that neither visual stimulation nor locomotion significantly affected the proportion of compartment-specific events; the function relating calcium transient amplitude and the percentage of attenuated events from soma to apical tuft remained unchanged by visual stimulation and behavioural-state changes (new panel B of Figure 5 and new Figure 5—figure supplement 1). Our results show that, on average, the percentage of coincident calcium signals across compartments of individual neurons was not significantly modified either by visual stimulation or behavioural state changes (n=19 neurons) (new Figure 5—figure supplement 1B and Figure 5—figure supplement 2) (subsection “Visual stimulation and locomotion do not alter the coupling of calcium signals between neuronal compartments from soma to apical tuft”).

We have added a full new figure showing how the interaction of locomotion and visual stimulation with drifting gratings affects global dendritic calcium signals and somatic responses (Figure 6). We have analysed the responses of 15 orientation-selective neurons. These results show that locomotion increased calcium transient amplitudes both in apical tuft and soma, without a significant difference between compartments (Figure 6G). This gain in calcium transient amplitude was not significantly different during the presentation of preferred and non-preferred orientations (Figure 6H). As a consequence, the orientation selectivity index (OSI) of somatic and apical tuft responses remained unchanged during both stationary and locomotion periods (Figure 6I). In line with these results, we found that the preferred orientation was similar throughout tuft dendritic branches and neuronal compartments of individual neurons, both during stationary and locomotion periods (Figure 6J and K). Finally, the Pearson’s correlation between apical tuft branches calcium signals and across compartments remained high and unchanged regardless of the gratings orientation and behavioural state (still or locomotion) (Figure 6L, 6M) (subsection “Visual stimulation and locomotion do not alter the coupling of calcium signals between neuronal compartments from soma to apical tuft” paragraph three).

Consequently, based on the repertoire of behavioural states and visual stimuli assessed in this study, we did not find a significant impact of behavioural events or stimuli on the proportion of somatic signals that fail to invade the distal dendrite. However, this does not rule out the influence of untracked behavioural and cognitive states that may differentially affect somato-dendritic coupling.

[…]Essential Revisions:1) The observation that behavior or visual stimulation did not influence the coupling between distal tuft and soma may seem intriguing. But the current behavior paradigm does not fully engage the animal in a well-controlled task. There could be untracked behavioral events other than locomotion, and there could be covert cognitive processes going on even in the darkness without running. These untracked behavioral events or cognitive processes may produce global dendritic spikes. Even without locomotion and visual stimulation, there can still be hidden variables that would produce local dendritic signals that cannot be isolated. The authors should provide a more comprehensive and cautious interpretation to their observations.

We fully agree and have edited the text accordingly.

2) The analysis of dendritic calcium signals is relatively crude, regarding their relationship to behavioral parameters. The two most important variables in the behavioral paradigm are the visual stimulation and locomotion. When studying dendritic integration, an obvious key question is how dendrites in the large L5 neurons integrate these two inputs. How might locomotion influence visual responses in the dendrites? Would there be a gain modulation of visual responses by locomotion, and if so, could such gain modulation be mediated by dendritic integration? The authors should at least quantify the relationship between dendritic signals and locomotion, and compare dendritic responses to the same visual stimuli between stationary and locomotive state. The authors showed orientation tuning of dendritic responses and its relationship to correlations, which is nice. But how would locomotion influence such orientation tuning and correlations? Currently there is only Figure 1H showing the mean fluorescence changes under different conditions, which is not very informative.

We thank the reviewers for these suggestions. We have added a full new figure to compare somatic and dendritic responses to drifting gratings between stationary and locomotion periods (new Figure 6) (subsection “Visual stimulation and locomotion do not alter the coupling of calcium signals between neuronal compartments from soma to apical tuft” paragraph three).

3) It's not clear what the biophysical interpretation of the asymmetric coupling is. Electrically, in a passive dendritic tree, proximal to distal propagation is more efficient than distal to proximal propagation. Here the data show an asymmetry of calcium transients in the reverse direction. In principle this could be due to distal calcium transients being associated with large dendritic spikes, which would always trigger somatic action potentials and thus an apparent strong "coupling", while proximal transients might include action potentials that do not backpropagate fully into the apical dendrite. However, looking at the data in Figure 3, there are examples of proximal/distal transient pairs that are smaller than large proximal transients that don't show up in the apical dendrites. What is the authors interpretation? Is the asymmetry coming from inhibition?

We have performed ex vivoexperiments presented in the new Figure 4 to investigate the biophysical interpretation of the asymmetric coupling found in vivo. These findings are described in a new result section (subsection “Ex vivo calibration of GCaMP6s and GCaMP6f signals in layer 5 soma and apical tuft dendrites”). Our findings from ex vivoexperiments suggest that: 1) somatically-triggered back-propagating action potentials below the critical frequency (<50-100Hz) may underlie the distant-dependent loss of dendritic calcium signals we observed in vivo; 2) spiking driven by strong apical tuft input or high-frequency spiking driven by somatic depolarization may underlie the strong somatic-dendritic coupling detected in vivo.

We thank the reviewers to point out the important observation that some relatively large calcium transients in soma and proximal trunk were not detected in the distal compartments (see examples in Figure 3B(iii) and 3B(iv) and percentage of compartment-specific event in Figure 3C(iii) and 3C(iv) for ΔF/F0>0.3), suggesting the presence of active mechanisms inhibiting calcium transients along the apical trunk. We have added a description of these results (subsection “Frequency of calcium transients decreases in a distance- and amplitude dependent manner from soma to apical tuft” paragraph two) and a Discussion of potential active mechanisms, including inhibition and neuromodulation (subsection “High but asymmetric coupling of calcium signals from soma to apical tuft”).

4) The authors should provide some measure of the relationship between calcium and voltage by performing whole-cell recordings from soma and dendrites of GCamp6-expressing L5 cells in vitro, as Beaulieu-Laroche et al. have done. This is particularly important because the asymmetry in the somato-dendritic coupling reported here contradicts the findings of Beaulieu-Laroche et al. While the difference could be due to the indicator, it would be important to experimentally clarify where the difference comes from.

We have performed ex vivoexperiments by combining calcium imaging and whole-cell recordings in acute cortical slices from mouse V1. We imaged the soma and distal apical dendrite (nexus) of GCaMP6s- and GCaMP6f- expressing layer 5 neurons whilst driving neuronal spiking either by somatic current injections (somatic stimulation) or L1 electrical stimulation (dendritic stimulation) (new Figure 4). These findings are described in a new result section (“Ex vivo calibration of GCaMP6s and GCaMP6f signals in layer 5 soma and apical tuft dendrites”).

These results show that, compared to GCaMP6s, GCaMP6f had a reduced sensitivity for detecting calcium signals, specifically in the soma during low frequency spiking driven either by somatic or dendritic stimulation (Figure 4E-4H). Consequently, as compared to GCaMP6s, attenuation of events from soma to dendrite was underestimated whereas attenuation of events from dendrite to soma was overestimated with GCaMP6f (Figure 4G vs 4H). With GCaMP6s, by contrast, attenuation of signals was asymmetric; we only found attenuation from soma to distal dendrites, as observed in vivo(Figure 4G). This difference between GcaMP6s and GCamP6f is consistent with the observation of a symmetric somato-dendritic coupling in the study of (Beaulieu-Laroche et al., 2019) using GCaMP6f and the observation of an asymmetric coupling in our results, using GCaMP6s (paragraph four of subsection “Ex vivo calibration of GCaMP6s and GCaMP6f signals in layer 5 soma and apical tuft dendrites”). We added a discussion about the difference between Beaulieu-Laroche et al., and our findings (subsection “High but asymmetric coupling of calcium signals from soma to apical tuft” paragraph two).

5) The conclusion about the correlation between distal dendrites and soma is based on indirect measurements. In Figure 2, the authors showed the correlations between compartments of neighboring focal planes. But presumably due to the technical limitation, there is no direct measurement of the correlation between the distal tuft branches and somatic signals. The author should at least provide certain estimation about to what extent the multiple correlations between neighboring compartments may tell about the direct correlation between distal branches and soma.

It is correct that due to technical limitations of the piezoelectric device we did not simultaneously image the soma and corresponding apical tuft. Pairs of compartments (soma-trunk, trunk-trunk, trunk-tuft, tuft-tuft) were imaged at a distance of 170 μm. We now provide a detailed explanation of our estimation of the proportion of compartment-specific events from soma to apical tuft (Figure 3—figure supplement 1): we estimated a decrease of about 40% of calcium transients from soma to the distal part of the apical tuft. This quantification was confirmed by a second data set, in which somatic and apical tuft calcium transients were imaged independently in individual layer 5 neurons. We found that the mean frequency of calcium transients in the apical tuft corresponded to 62% of the frequency of events in the corresponding soma (n = 13 neurons, Figure 3—figure supplement 1). We have edited the text accordingly (subsection “Frequency of calcium transients decreases in a distance- and amplitude

dependent manner from soma to apical tuft” and Figure legend of Figure 3—figure supplement 1).

6) Although the authors used CaMKII to restrict the expression of Cre in excitatory neurons, this study did not distinguish different subtypes of L5 pyramidal neurons. It is known that there are at least two or more different subtypes of L5 neurons with distinct dendritic morphologies, dendritic integrative properties, and downstream projection patterns. For example, the cortico-cortical projection L5 neurons are mostly thin tufted neurons, showing less bursting activity, which may be attributable to differences in the properties of dendritic spikes. Additionally, the distal tuft of the thick tufted neurons may be more compartmentalized than the thin tufted neurons, and show large dendritic calcium spikes. How such differences in different subtypes of L5 neurons may influence the observations in the current study? At least, the authors should include a discussion on this issue.

We thank the reviewers for this suggestion. We have added a discussion of how both apical tuft signals and the somato-dendritic coupling may differ between different subtypes of layer 5 pyramidal neurons (subsection “High but asymmetric coupling of calcium signals from soma to apical tuft” paragraph four). The neurons included in this study had their soma located at various depths within layer 5 (median 528 μm, see Figure 3—figure supplement 1) and mainly displayed thick-tufted morphology, characteristic of pyramidal tract neurons. Since we selected the imaged neurons visually based on their GCaMP6 fluorescent signal, our sampled is likely biased towards layer 5 neurons with thick trunk and thick-tufted morphology.

[Editors' note: further revisions were requested prior to acceptance, as described below.]

The revised manuscript has been significantly improved with inclusion of additional experiments and analysis and careful editing of the text. However, there is a remaining issue that needs to be addressed prior to further consideration of the manuscript. This requires an additional set of ex vivo experiments that is crucial to the main conclusions of the paper, which we think are feasible to complete within a month.The reviewers have pointed out the shortcomings related to the new ex vivo experiments (Figure 4). Specifically, the use of suprathreshold dendritic stimulation that also evokes somatic APs strongly activates both dendrites and soma, and produces an apparent strong coupling in the dendrite to soma direction. However, given that dendritic stimulation can evoke somatic APs, bpAPs could have interacted with local dendritic activity. Therefore, it is not clear if in the present imaging conditions, how local dendritic events lead to global dendritic spikes and if there are dendritic events that do not cause the soma to fire. In order to support the claim of the rarity of such events in vivo, or to test for the possibility that they cannot be detected, the authors need to show by imaging using a range of subthreshold dendritic stimulations given to layer 1 in the ex vivo preparation, the dendrite to soma coupling as a function of the level of dendritic calcium activity.

As suggested, we have now included a new supplementary figure that shows the dendrite to soma coupling for a range of subthreshold dendritic stimulations given to layer 1, in our ex vivopreparation (Figure 4—figure supplement 1). We have shown this both as a function of somatic output (Figure 4—figure supplement 1D, E), and as a function of the level of dendritic calcium activity (Figure 4—figure supplement 1F).

Specifically, we present data in which we image GCaMP6s in both the soma and dendrite while delivering stimulation to L1 at varying frequencies (5-200Hz) and with intensities at 25%, 50%, 75% and 100% of action potential threshold. Our principle findings are:

1) We could not detect any global calcium signals in the apical tuft when L1 stimulation failed to evoke a somatic action potential. Thus, there is a strong coupling between global calcium events and somatic spiking in our experimental conditions. This is largely consistent with several studies demonstrating that apical tuft electrogenesis is associated with somatic spiking (*e.g.* Helmchen et al., 1999, Larkum and Zhu, 2002, also Beaulieu-Laroche et al., 2019).

2) Provided L1 stimulation evoked at least 2 action potentials, we could detect calcium signals in both apical tuft and soma using GCaMP6s. However, when L1 stimulation only evoked a single action potential, apical tuft calcium signals were still observed in 60% of the trials; however, these events were now below the detection limit of GCaMP6s in the soma. Consequently, under these conditions, soma-dendrite coupling was low.

These data reveal that evoked apical tuft calcium events are strongly coupled to somatic spiking, and that this coupling can be detected by GCaMP6s imaging providing that somatic output is above the limits of detection (>2 action potentials ex vivo). We have edited the manuscript accordingly (Results subsection “Ex vivo calibration of GCaMP6s and GCaMP6f signals in layer 5 soma and apical tuft dendrites” paragraph 5; Materials and methods subsection “Ex vivo patch-clamp recordings and calcium imaging”).

We have also added to the discussion (in the section “Limitations of the use of calcium imaging to assess dendritic activity in awake behaving mice”), a paragraph about the limitations of detecting branch-specific events in vivo. Whereas previous ex vivo studies have demonstrated that branch-specific stimulation can indeed trigger localized Ca^2+^ events detected by calcium indicators (e.g. Cai et al., 2004; Sandler et al., 2016; review: Major et al., 2013), the detection of these events in vivo is more challenging, principally because they may be masked by ongoing global dendritic events. Given the relatively low temporal resolution of GCaMP6s, dissociating global dendritic spikes from potential underlying branch-specific events is challenging (Kerlin et al., 2019). The contribution of branch-specific calcium signalling to ongoing neuronal activity in vivo therefore remains an open question.

References:

Cai X, Liang CW, Muralidharan S, Kao JP, Tang CM, Thompson SM. 2004. Unique roles of SK and Kv4.2 potassium channels in dendritic integration. Neuron 44:351-64

Helmchen, F., Svoboda, K., Denk W., Tank D.W., (1999) in vivo dendritic calcium dynamics in deep-layer cortical pyramidal neurons, Nature Neuroscience. doi: 10.1038/14788.

Kerlin, A. et al. (2019) ‘Functional clustering of dendritic activity during decision-making’, *ELife*. Oct 30;8. pii: e46966. doi: 10.7554/*eLife*.46966.

Larkum ME, Zhu JJ. 2002. Signaling of layer 1 and whisker-evoked Ca^2+^ and Na^+^ action potentials in distal and terminal dendrites of rat neocortical pyramidal neurons in vitro and in vivo. J. Neurosci. 22:6991-7005

Major G, Larkum ME, Schiller J. Active properties of neocortical pyramidal neuron dendrites, Annu Rev Neurosci. 2013 Jul 8;36:1-24.

Sandler M, Shulman Y, Schiller J. A Novel Form of Local Plasticity in Tuft Dendrites of Neocortical Somatosensory Layer 5 Pyramidal Neurons. Neuron. 2016 Jun 1;90(5):1028-42.